# Intradermal but not intramuscular modified vaccinia Ankara immunizations protect against intravaginal tier2 simian-human immunodeficiency virus challenges in female macaques

Venkata S. Bollimpelli[1], Pradeep B. J Reddy[1], Sailaja Gangadhara[1], Tysheena P. Charles[1], Samantha L. Burton[1], Gregory K. Tharp [2], Tiffany M. Styles[1], Celia C. Labranche [3], Justin C. Smith [4], Amit A. Upadhyay [1], Anusmita Sahoo[1], Traci Legere[1], Ayalnesh Shiferaw[1], Vijayakumar Velu [1,5], Tianwei Yu[6], Mark Tomai[7], John Vasilakos[8], Sudhir P. Kasturi[5], George M. Shaw[9], David Montefiori [3], Steven E. Bosinger [5], Pamela A. Kozlowski [4], Bali Pulendran [10], Cynthia A. Derdeyn [5,12], Eric Hunter [5] & Rama R. Amara [1,11] ✉

Route of immunization can markedly influence the quality of immune response. Here, we show that intradermal (ID) but not intramuscular (IM) modified vaccinia Ankara (MVA) vaccinations provide protection from acquisition of intravaginal tier2 simian-human immunodeficiency virus (SHIV) challenges in female macaques. Both routes of vaccination induce comparable levels of serum IgG with neutralizing and non-neutralizing activities. The protection in MVA-ID group correlates positively with serum neutralizing and antibody-dependent phagocytic activities, and envelope-specific vaginal IgA; while the limited protection in MVA-IM group correlates only with serum neutralizing activity. MVA-ID immunizations induce greater germinal center Tfh and B cell responses, reduced the ratio of Th1 to Tfh cells in blood and showed lower activation of intermediate monocytes and inflammasome compared to MVA-IM immunizations. This lower innate activation correlates negatively with induction of Tfh responses. These data demonstrate that the MVA-ID vaccinations protect against intravaginal SHIV challenges by modulating the innate and T helper responses.

Multiple vaccine strategies are being pursued to achieve the formidable goal of preventing HIV infection. One of the primary goals of many of these approaches is to induce high titer and long-lasting neutralizing antibodies. In this direction, studies in rhesus macaques using pre-fusion stabilized native-like HIV envelope (Env) protein immunogens delivered with potent adjuvants have been shown to induce a strong autologous neutralizing antibody response[1–5]. Recently, we and others showed that for protein-only

vaccines, which primarily induce antibody response with little or no CD8 T cell response, an autologous serum neutralization titer of about 300 or greater is required for protection against mucosal SHIV BG505 challenges in rhesus macaques[2,5]. The vaccine-induced antibodies showed limited or no breadth of neutralizing activity against heterologous tier2 Env, and the protection was absent in animals with low neutralizing antibody titer. To test the synergy between neutralizing antibodies and tissue-resident T cells, we combined BG505 SOSIP.664 trimer protein vaccinations with heterologous viral vector (HVV) vaccine boosts expressing SIV Gag to induce strong Gag-specific CD8 T cell response[5]. The HVV vaccination consisted of sequential immunizations with recombinant vesicular stomatitis virus (VSV), vaccinia virus (VV), and adenovirus type 5 (Ad5) vectors expressing SIV239 Gag administered intravenously to induce strong tissue-resident memory T cells. Impressively, this protein+HVV vaccine strategy demonstrated protection even in animals with a neutralizing antibody titer of less than 300. The results from this protein+HVV vaccine strategy served as a proof of concept in demonstrating the beneficial effects of combining antibody-inducing vaccines with T cell-inducing vaccines to enhance protection against HIV. The results also highlighted the need for the development of such approaches using vaccine vectors and different immunization routes with translational potential.

The heterologous DNA prime and modified vaccinia virus Ankara (MVA) boost approach (DNA/MVA) has been shown to be safe and immunogenic to induce strong CD4 and CD8 T cell responses in animal models and humans[6–14]. Our DNA and MVA vaccines are designed to express the trimeric form of HIV-Env displayed on virus-like particles (VLPs) that allow the presentation of Env in its native state. Our previous study showed that the addition of protein boosts to DNA/MVA vaccination markedly boosts the antibody response[15]. Based on these findings, in the current study, we used DNA/MVA/Protein vaccine regimen, where we included pre-fusion stabilized native-like BG505 SOSIP protein boosts adjuvanted with 3M-052, a TLR7/8 agonist, encapsulated in poly-lactic glycolic acid (PLGA) nanoparticles (3M-052-NP). The BG505 SOSIP.664 trimer protein vaccination with 3M-052-NP adjuvant has been shown to induce a strong autologous neutralizing antibody response in macaques[5]. This approach also allowed us to test if priming with trimeric HIV-Env presented on VLPs followed by boosting with native-like trimeric HIV-Env protein boost can further enhance the generation of neutralizing antibody response.

The route of vaccine administration can influence vaccine efficacy by modulating the magnitude, functional quality, and tissue localization of the vaccine-induced immune responses[16–18]. The composition of antigen-presenting cells is quite distinct in the dermis and muscle[19–21], with the former but not the latter rich in Langerhans cells, which are potent antigen-presenting cells. Thus, intradermal (ID) vaccinations target the antigen to different antigen-presenting cells compared to intramuscular (IM) vaccinations, which might induce different T cell and antibody responses. Hence, in the current study, we compared the immunogenicity and efficacy of ID and IM routes of MVA vaccination as part of a DNA/MVA/Protein vaccine regimen for protection against repeat intravaginal tier2 SHIV challenges in rhesus macaques. Our results showed that both MVA-ID and MVA-IM immunizations induce strong and comparable T cell and antibody responses. However, they differed in the quality of immune responses they generated, with the MVA-ID vaccination inducing more of a Tfh response and lower activation of intermediate monocytes compared to the MVA-IM vaccine. Furthermore, MVA-ID vaccination but not MVA-IM vaccination provided significant protection against the SHIV challenge. While animals in both the vaccine groups displayed strong control of viral replication following breakthrough infection, it was significantly better in the MVA-IM group compared MVA-ID group.

## Results

### MVA-ID but not MVA-IM vaccinations protect against acquisition of intravaginal tier2 SHIV challenges

We vaccinated two groups of rhesus macaques with 10 animals per group using the DNA/MVA/Protein vaccine regimen (Fig. 1a). Animals in both groups were primed with DNA/SHIV BG505-CD40L via the ID route and electroporation at weeks 0 and 8. At weeks 16 and 24, all animals were boosted with MVA/SHIV BG505 delivered through either ID or IM routes. Both DNA and MVA immunogens expressed SIVmac239 Gag, protease, and reverse transcriptase; Clade A BG505 Env; and Clade D Tat and Rev (SFig. 1). In addition, DNA co-expressed CD40L as an adjuvant to provide a costimulatory activity that can enhance both humoral and cell-mediated immune responses[22]. While the DNA expressed full-length Env protein gp160, MVA expressed a C-terminally truncated Env protein gp147 (SFig. 2). Later, both groups were boosted with BG505 SOSIP.664 protein adjuvanted with nanoparticle encapsulated TLR7/8 agonist 3M-052 (NP-3M-052) delivered subcutaneously at weeks 32, 48, and 66. A control group with 15 animals received one dose of NP-3M-052 subcutaneously 4 weeks prior to the SHIV challenge. Four weeks after the last immunization, all animals were challenged intravaginally with tier2 BG505.332 N.375YDCT SHIV virus on a weekly basis (Fig. 1a). The weekly challenges continued until the animal became positive for the virus in plasma for two consecutive weeks or for a maximum of ten exposures. As can be seen in Fig. 1b, the strong protection against intravaginal SHIV challenges in the DNA/MVA/Protein vaccinated animals was observed only in the MVA-ID group and not in the MVA-IM group. The difference in the rate of acquisition was striking, and we observed that 80% of control and 70% of MVA-IM animals were infected by the fourth challenge, while only 30% of the MVA-ID animals were infected at this point. By the end of ten challenges, 40% of MVA-ID group animals remained uninfected with an estimated 73% vaccine protection/exposure ($P = 0.006$). On the other hand, animals in the MVA-IM group did not show any significant delay in viral acquisition compared to controls (Fig. 1b).

We monitored plasma viremia until 11 weeks in animals that showed breakthrough infection. Post-productive infection, animals in both vaccinated groups showed a 2-log reduction in peak viremia compared to controls, and the viral control was better in the MVA-IM group compared to the MVA-ID group. At 11 weeks post-infection, the viral control was significantly better in the MVA-IM group compared to controls, although there was a trend for better viral control in the MVA-ID group as well (Fig. 1c). In addition, at week 3 post-infection, the viral control was greater in the MVA-IM group compared to MVA-ID group ($p = 0.008$). The area under the curve analysis of viral load from weeks 0 to 11 also indicated better control of viremia in both vaccine groups compared to controls, and superior viral control in the MVA-IM group compared to the MVA-ID group (Fig. 1d). These viral load kinetics in infected animals was similar irrespective of the challenge number they got infected (SFig. 3). These results demonstrated that the DNA/MVA/Protein vaccination protects against a homologous tier2 intravaginal SHIV challenge and the route of MVA administration can profoundly alter the protection outcome with MVA-ID favoring the protection from the acquisition of infection and the MVA-IM favoring enhanced viral control. These results also suggested a differential vaccine-induced immune response between the two vaccine groups.

### Immune correlates of protection differ by the route of MVA vaccination

All vaccinated animals showed a strong binding antibody response in serum against the BG505 SOSIP gp140 trimer and these responses were largely comparable between the groups, with small differences at a few time points (Fig. 2a). The binding antibody response was evident right after the first MVA vaccination, where the titer was sixfold higher in the MVA-IM group (geometric mean of 33 ug/mL) compared to MVA-ID group (geometric mean of 5 mg/mL). This

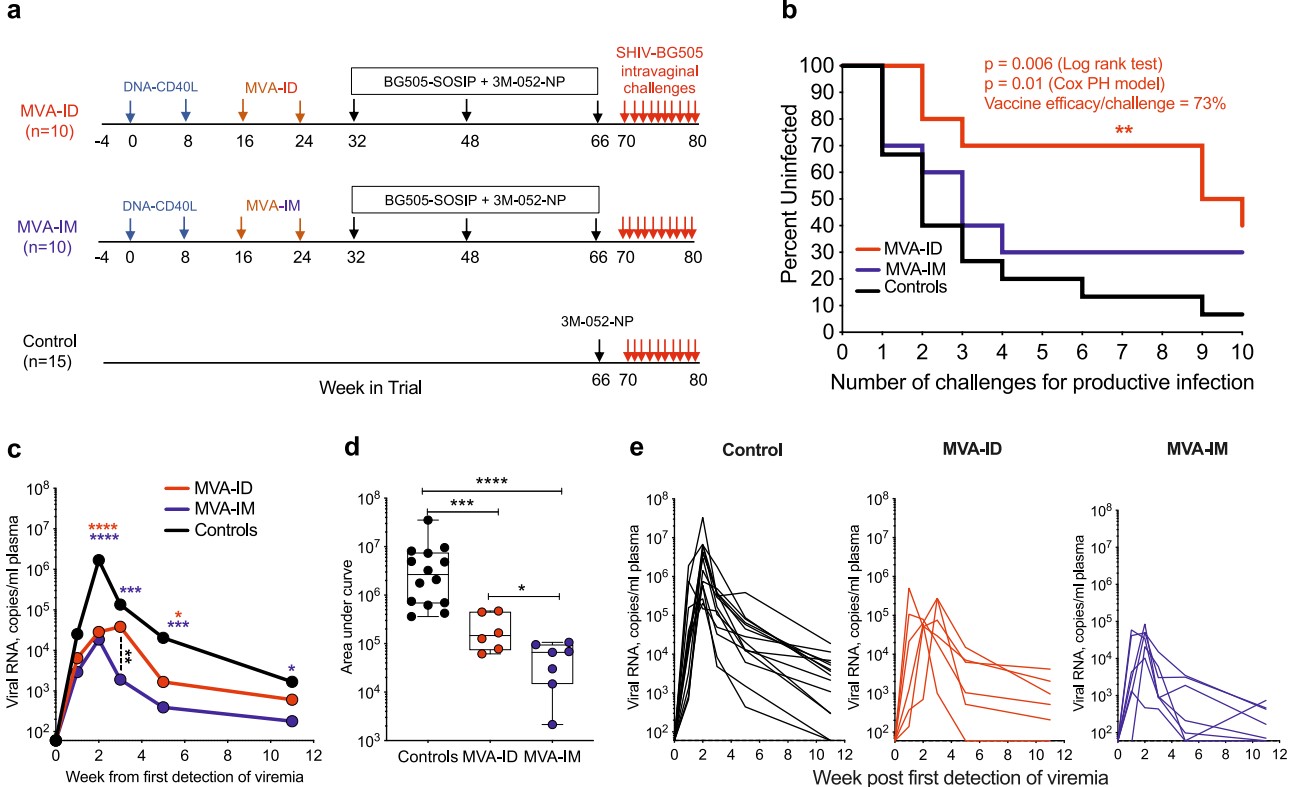

**Fig. 1 | DNA/MVA-ID/protein vaccine shows protection against acquisition of infection on intravaginal low dose SHIV BG505 challenge. a** Schematic representation of vaccine groups and the immunization schedule. Arrows were color-coded to represent the immunogen/virus challenge and denote the time in weeks of their administration. **b** Kaplan–Meier survival curves representing the fraction of uninfected animals following each challenge on the y-axis ($n = 10$ in each vaccinated group and $n = 15$ in the control (adjuvant-only) group). **c–e** Plasma viral load in infected animals post-infection. The week in which viremia was first detected was considered as Wk1. **c** Longitudinal viral load profile representing the geomean of all infected animals in each group. Asterisks denoting statistical significance measured by Mann–Whitney rank-sum test (two-tailed) and were color-coded for each group (red: MVA-ID; blue: MVA-IM), compared to control group, and black: MVA-ID

compared to MVA-IM group; second week: red/blue: ****$p < 0.0001$; third week: blue: ***$p = 0.0001$, black: **$p = 0.008$; fifth week: red: *$p = 0.02$, blue: ***$p = 0.0005$; 11th week: blue: *$p = 0.018$. **d** Area under the curve of longitudinal viral load in infected animals, compared between the groups; box in the plot extends from the 25th percentile to the 75th percentile of the dataset and the line inside the box denotes the median. The whiskers descend to the dataset's minimum values and ascend to the maximum values; Asterisks in viral load data, denote statistically significant differences between the groups at the given time point, measured by Mann–Whitney rank-sum test (two-tailed) (****$p < 0.0001$, ***$p = 0.0006$, *$p = 0.035$). **e** Longitudinal viral load profile of each infected animal. Infected animals in control group: $n = 14$, MVA-ID group: $n = 6$, MVA-IM group: $n = 7$. Source data are provided as a Source Data file.

response was boosted by the second MVA vaccination and was comparable between the groups at about 110 ug/mL. The antibody response was further boosted by each SOSIP protein immunization with a titer of about 618 ug/mL following the first protein vaccination in the MVA-ID group and 273 ug/mL in the MVA-IM group. Both groups reached ~1200 ug/mL by the day of the challenge (Fig. 2a). Vaccinations also induced high levels of SOSIP-specific IgG and IgA in vaginal secretions. Similar to responses in serum, the antibody responses in vaginal (Fig. 2b, c) and rectal (SFig. 4b, c) secretions also increased with each boost and peaked after the 2nd protein boost with no further increase after the third protein boost.

In contrast to the binding antibody response, the neutralizing antibody response was not evident until after the first protein immunization, where a low level of autologous neutralizing antibody response was detected in 5 out of the 20 animals. The number of responders increased to 12 following the second protein immunization. Earlier responders of the first protein immunization showed enhanced titers with the second protein boost and one animal in the MVA-ID group reached a very high ID$_{50}$ titer of ~1:6000, following Protein-3 immunization. A third protein immunization did not boost the neutralization titer significantly, as only a few animals in the MVA-ID group showed enhancement of titers, while other animals from both groups did not show any enhancement. On the day of the challenge, 11 out of 20 animals maintained a neutralization titer above the detection

limit. Overall, neutralization titers were quite variable within each group and were comparable between the groups (Fig. 2d).

Neutralizing antibody titers on the day of the challenge showed a significant correlation with the rate of viral acquisition in both the vaccinated groups. However, this association was strikingly stronger in MVA-IM than the MVA-ID group (Fig. 2e), suggesting a higher dependence on neutralization antibodies for protection in MVA-IM than the MVA-ID group. Interestingly, vaccinated animals with neutralizing antibody titer >1: 50 remained uninfected (3/3 in MVA-IM, 2/3 in MVA-ID) or showed delayed acquisition (1/3 in MVA-ID, infected at tenth challenge) (Fig. 2f). However, in animals with a neutralization titer <1:50, protection was observed only in the MVA-ID group. There was a significant difference in the acquisition of infection between the two vaccine groups among the animals with <1:50 neutralization titer ($p = 0.023$). All seven of these animals with low neutralization titers in the MVA-IM group showed an early acquisition of infection (infected within four challenges). In the case of the MVA-ID group, out of seven low-neutralizing titer animals, two remained protected, two showed delayed acquisition (infected at the ninth challenge), and the other three showed early infection (infected within four challenges). Even in these animals with weak neutralizing antibody response, the MVA-ID group showed a trend towards delayed virus acquisition compared to the control group (Fig. 2g). This observation suggests that other immune parameters

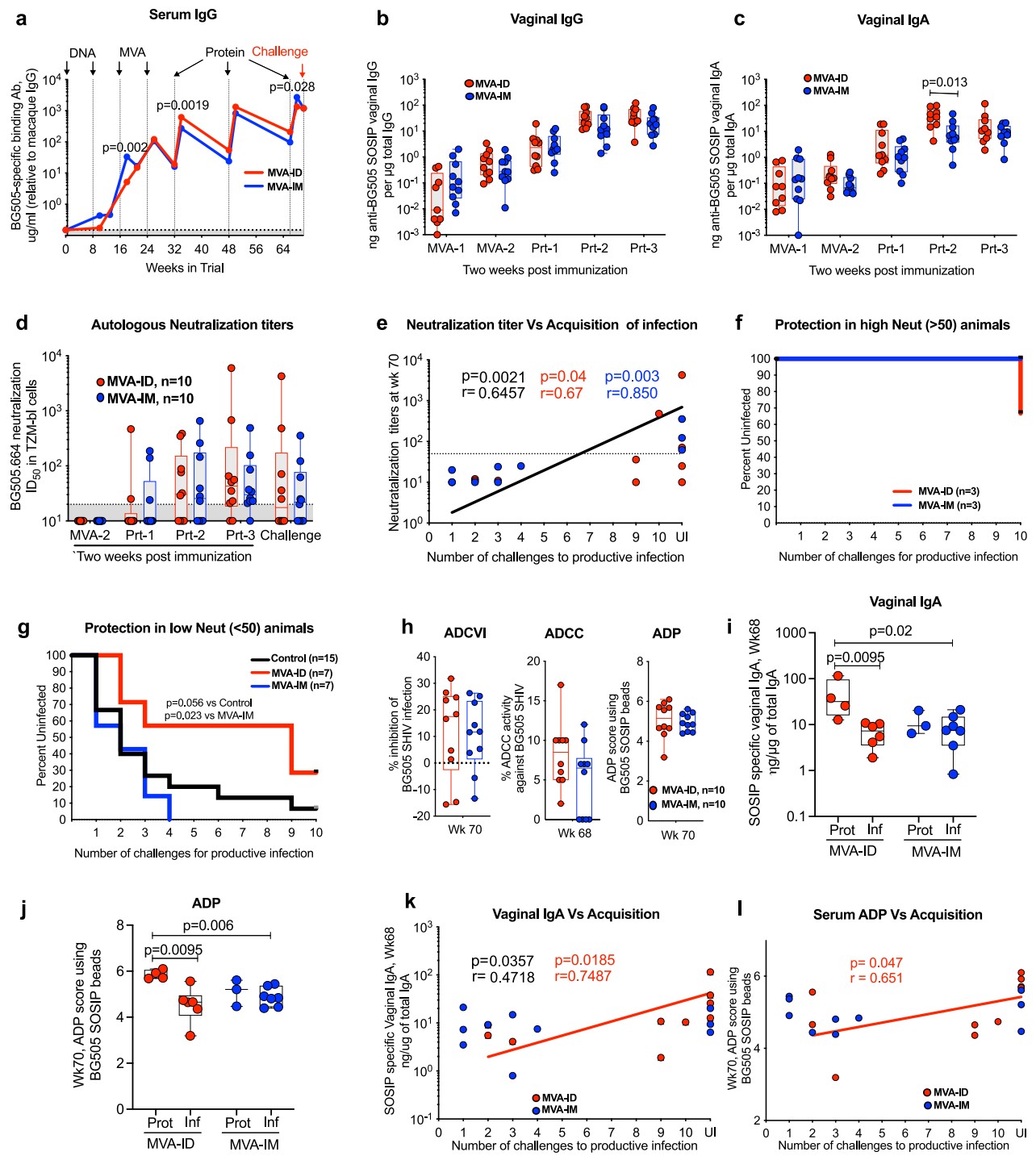

**Fig. 2 | MVA-ID vaccinated animals show different correlates of protection compared to MVA-IM vaccinated animals. a** BG505_SOSIP-specific IgG in serum. Statistical significance by Mann–Whitney rank-sum (two-tailed) test. **b**, **c** BG505_SOSIP-specific IgG and IgA in vaginal secretions; Vaginal IgG: (MVA-ID: $n = 9$ at MVA-1 and $n = 10$ at rest of the time points; MVA-IM: $n = 10$ at all time points); Vaginal IgA: (MVA-ID: $n = 9$ at MVA-1 and Prt-2 time ponts and $n = 10$ at rest of the time points; MVA-IM: $n = 10$ at all time points). Statistical significance by Mann–Whitney rank-sum (two-tailed) test. **d** Neutralization antibody $ID_{50}$ titers against BG505.T332N Env pseudovirus in serum. **e** Spearman's correlation (two-sided) between neutralizing antibody titers and rate of virus acquisition. The dotted line represents the neutralization titer at 50. **f**, **g** Kaplan–Meier survival curves in MVA-ID and MVA-IM groups with **f** high neutralizing antibody titer and **g** low-neutralizing antibody titer, compared to control animals. Statistical significance by log-rank test. **h** Serum antibody effector functions compared between MVA-ID and MVA-IM groups. Left to right: ADCVI inhibition, ADCC activity, ADP score. **i** Vaginal

IgA and **j** serum ADP score in protected and infected animals within a group and compared between both vaccinated groups. Statistical significance by Mann–Whitney rank-sum (two-tailed) test; $n = 4$ in MVA-ID protected; $n = 6$ in MVA-ID infected; $n = 3$ in MVA-IM protected; $n = 7$ MVA-IM infected. Spearman's correlation (two-sided) between rate of virus acquisition and **k** anti-BG505 SOSIP vaginal IgA, **l** serum ADP score. In all the correlation plots, $p$ and $r$ values are color-coded to represent the vaccine group (red: MVA-ID, Blue: MVA-IM, Black: both groups combined). These values were not shown for the group that did not have a statistically significant $p$ value <0.05. Boxes in all the box plots extend from the 25th percentile to the 75th percentile of the dataset and the line inside the box denotes the median. The whiskers outside the box descend to the dataset's minimum values and ascend to the maximum values. MVA-1: first MVA; MVA-2: second MVA; Prt-1: first protein; Prt-2: second protein; Prt-3: third protein, UI uninfected. Source data are provided as a Source Data file.

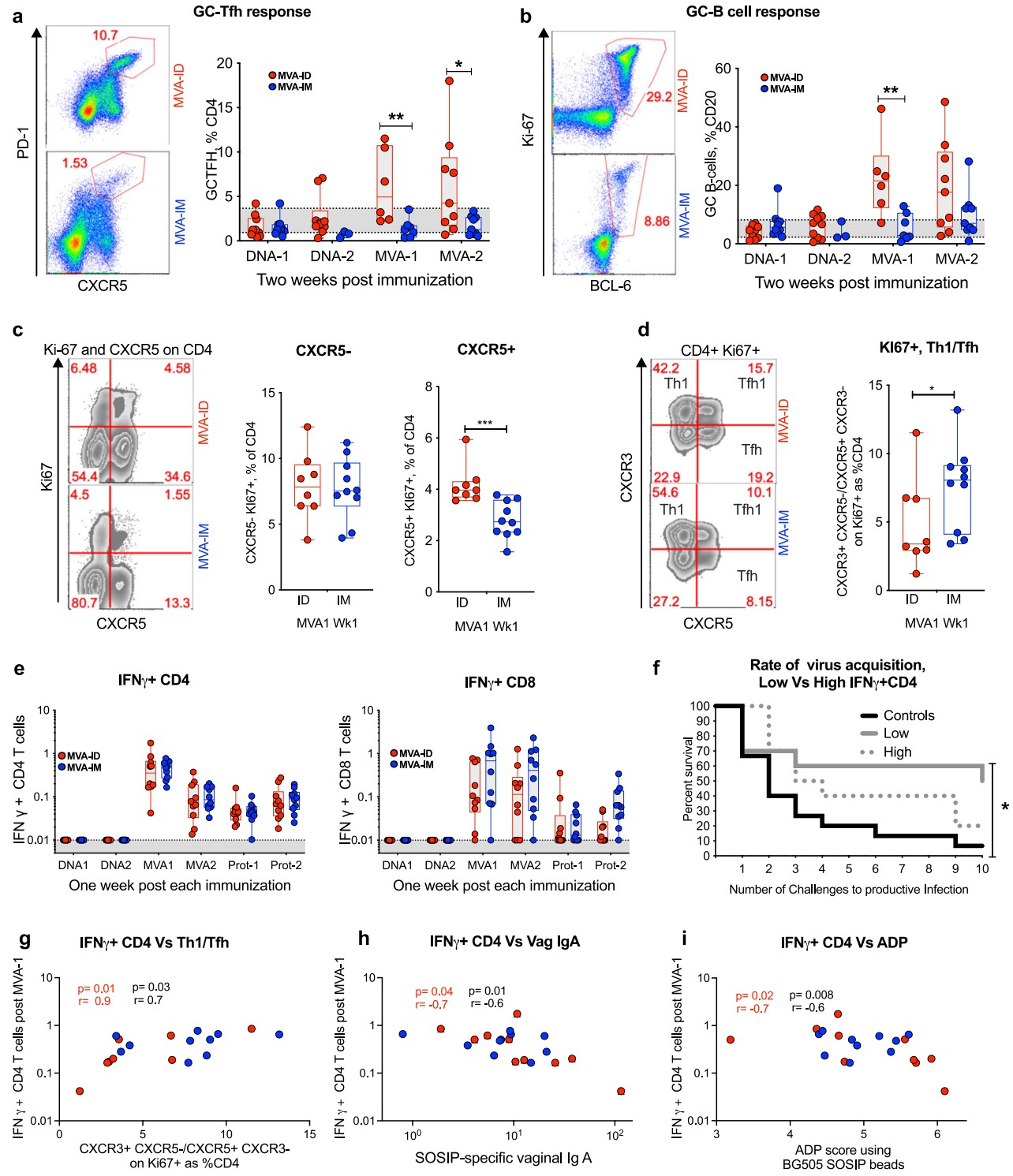

besides neutralizing activity contributed to protection in the MVA-ID group.

To address this, we measured antibody-mediated effector functions such as antibody-dependent cell-mediated viral inhibition (ADCVI), antibody-dependent cellular cytotoxicity (ADCC), and antibody-dependent cellular phagocytosis (ADCP) and found that they all were comparable between the groups (Fig. 2h). However, we found that vaginal IgA response and serum ADP score were significantly higher in protected animals of MVA-ID group compared to infected animals in both MVA-ID and MVA-IM groups (Fig. 2i, j).

Further, vaginal IgA and serum ADP scores showed a significant positive association with delay in virus acquisition in the MVA-ID group but not in the MVA-IM group (Fig. 2k, l). Post challenge, peak viral load in infected animals showed a trend of negative correlation with peak (wk68, 2 weeks post third protein) serum BG505-specific IgG binding antibody (SFig. 5a). These observations demonstrated strong neutralizing antibody response contributing to protection in both vaccine groups and suggested a role for non-neutralizing serum antibodies and vaginal IgA for protection only in the MVA-ID group.

**Fig. 3 | T cell response post-MVA immunization. a** Germinal center Tfh response (CXCR5++PD1++) (at MVA-1**$p = 0.008$, at MVA-2 *$p = 0.031$) and **b** germinal center B-cell response (BCL6 + Ki67+) (**$p = 0.008$) measured on CD4 + T cells and CD20 + B cells respectively in FNAs processed draining inguinal lymph node. The shaded area represents the range of germinal center responses at pre-vaccination time points from MVA-ID animals. Data were not available for the MVA-IM group. Left: representative flow-cytometer plots post-MVA immunization. **c** Circulating non-Tfh response (CXCR5-Ki67+) and Tfh response (CXCR5 + Ki67+) (***$p = 0.0007$) measured on blood CD4 + T cells (frozen PBMC) post-MVA-1 immunization. Left: representative flow-cytometer plots. **d** Th1(CXCR5- CXCR3+) / Tfh(CXCR5 + CXCR3-) in blood Ki67 + CD4 + T cells (frozen PBMC) (*$p = 0.034$). Left: Representative flow cytometer plots. Asterisks in all the above data, denote statistically significant differences measured by Mann–Whitney rank-sum (two-tailed). **e** Total SHIV BG505 (Gag + Env) specific IFN-γ + CD4 T cell responses and CD8 T cell responses measured in blood. Boxes in all the box plots extend from the 25th percentile to the 75th percentile of the dataset and the line inside the box denotes the median. The whiskers outside the box descend to the dataset's minimum values and ascend to the maximum values. **f** Kaplan–Meier curves in animals with low and high frequencies of total SHIV BG505 (Gag + Env) specific IFN-γ + CD4 T cells (all vaccinated animals included and stratified as low and high, based on the median frequency as cut off) post-MVA-1 immunization compared with control animals. Statistical significance by log-rank test (*$p = 0.028$). **g–i** Spearman's correlation (two-sided) between Day 7 MVA-1 induced IFN-γ + CD4 T cells and **g** MVA-1, Day 7 Th1/Tfh ratio, **h** Wk68 vaginal IgA (ng anti-SOSIP IgA per μg total IgA), **i** Wk70, serum ADP score. In all the correlation plots, $p$ and $r$ values were color-coded to represent the vaccine group (red: MVA-ID, Blue: MVA-IM, Black: both groups combined). These values were not shown for the group that didn't have a statistically significant $p$ value <0.05. DNA-1: first DNA; DNA-2: second DNA; MVA-1: first MVA; MVA-2: second MVA; Prt-1: first protein; Prt-2: second protein. Source data are provided as a Source Data file.

## MVA-ID immunizations elicit higher Tfh response than MVA-IM immunization

We next focused on the influence of the route of immunization on T helper responses in the lymph node (LN) and blood, and germinal center (GC) B-cell responses in the LN (Fig. 3). We analyzed GC responses in the draining LN by evaluating fine-needle aspirates (FNAs) (Figs. 3a, b and SFig. 6a). We measured the frequency of germinal center T follicular helper cells (GC-Tfh) only in MVA-ID group prior to vaccination (baseline) and these cells ranged from 1–4% (gray shaded area on Fig. 3a). The frequency of these cells remained within the range of baseline levels after the two DNA vaccinations. However, following MVA vaccinations, we observed an increase in the MVA-ID group (average 6%) but not in the MVA-IM group (Fig. 3a). Similarly, the frequency of GC-B cells also increased by fourfold after the MVA boosts from a baseline of 5% to about 20% (Fig. 3b); again, only in the MVA-ID group. These results demonstrated that MVA-ID vaccinations are superior to MVA-IM vaccinations for inducing GC-Tfh and GC-B-cell responses.

We also studied the frequency of circulating Tfh responses based on CXCR5 and Ki67 expression on CD4 T cells in blood at one week after the first MVA boost. We chose this time point based on our previous study, where we showed that proliferating CXCR5+ and CXCR5-CD4 T cells peak at this time point following MVA-IM vaccination[23]. The frequency of CXCR5+ Ki67 + CD4 T cells was also higher in the MVA-ID group compared to the MVA-IM group (Fig. 3c and SFig. 6b). However, the frequency of CXCR5- Ki67+ cells was comparable between the two groups (Fig. 3c). Furthermore, the ratio of Th1 (CXCR3+) to Tfh (CXCR5+) cells among the proliferating CD4 T cells in the MVA-ID group was lower than MVA-IM group (Fig. 3d) suggesting a less Th1 biased response in MVA-ID than in MVA-IM group. The MVA vaccinations induced strong SHIV-specific (Gag+Env) CD4 and CD8 T cell responses, but these were comparable between the two groups. The IFNg+ CD4 T cell responses peaked after the 1st MVA vaccination and were not boosted further by the 2nd MVA vaccination. In contrast, the IFNg+ CD8 T cell response was comparable after both 1st and 2nd MVA immunizations. Protein immunizations did not boost IFNg+ T cell responses (Fig. 3e).

In our previous study, we showed a strong association between induction of strong IFNg+ CD4 T cell response (Th1 polarized potential target cells) following the first MVA vaccination, which represented the peak of the CD4 response, and diminished protection from the acquisition of infection[24]. To understand the association between SHIV-specific Th1 cells and protection in this study, we stratified vaccinated animals into two groups (low vs high) based on median IFNγ + CD4 T cell response and compared the rate of infection. Consistent with our earlier findings[24], animals with a lower vaccine-induced Th1 response showed a significant delay in the acquisition of infection (Fig. 3f). This detrimental effect of MVA vaccine-induced IFNg+ CD4 T cells was also observed post challenge, where these cells showed a positive correlation with peak viral load in infected animals (SFig. 5b). Interestingly, the MVA-induced IFNg+ CD4 T cells correlated positively with Th1/Tfh ratio (Fig. 3g) and was negatively associated with above correlates of protection like vaginal IgA (Fig. 3h) and serum ADP function (Fig. 3i). A comprehensive correlation matrix between protection correlates and other immune parameters is presented in SFig. 7. Collectively, these data demonstrated that changing the route of MVA administration from IM to ID skewed differentiation of Th response more toward Tfh and away from Th1 polarization, which could have improved the protective outcome of the vaccine by modulating the generation of IgA and ADP activity of the antibody response.

## MVA-ID immunization induces lower expansion and activation of intermediate monocytes compared to MVA-IM vaccination

The differential T cell response between the groups prompted us to look at the innate cell response post-MVA vaccination. We tracked the frequency and activation of monocytes after the first MVA immunization on Days 0, 1, 2, 4, and 7 in the blood. Monocytes were divided into three subsets based on the expression of CD14 and CD16 on HLADR+ cells, i.e., classical (CD14+, CD16−), intermediate (CD14+, CD16+), and non-classical (CD14−, CD16+) (Fig. 4a and SFig. 6c). The MVA vaccination rapidly increased the frequency of total and activated (CD86+) intermediate monocytes at Day 1 and this increase was significantly higher in the MVA-IM group compared to MVA-ID group (Fig. 4b, c). As expected, the increase in intermediate monocytes was associated with a decrease in classical monocytes on Day 1 post-vaccination, and this decrease was higher in the MVA-IM group compared to the MVA-ID group. The frequency of total and activated non-classical monocytes showed a small increase on Day 2 in the MVA-IM group but not in the MVA-ID group. Interestingly, the frequency of activated intermediate monocytes on Day 1 negatively correlated with the frequency of circulating Tfh (Fig. 4d), and positively correlated with Th1/Tfh ratio (Fig. 4E), IFNg+ CD4 T cells (Fig. 4F) on Day 7. These data demonstrated that the route of MVA vaccination could significantly alter the expansion and activation of intermediate monocytes and suggested a strong interplay between intermediate monocytes and circulating Tfh.

## MVA-ID vaccination induces lower AIM2/IFI16 mediated inflammasome activation that is associated with lower intermediate monocyte activation and stronger Tfh response

In an effort to find underlying innate mechanisms/pathways that are associated with the afore-described qualitative changes in the adaptive immunity following alternate MVA vaccination routes, we performed total blood RNA transcriptomic analysis at Days 0, 1, 2, 4, and 7 after the first MVA vaccination. Relative to Day 0, genes with expression $\log_2$fold change ≥1 or ≤−1 and $p$ value ≤0.05 were defined as differentially expressed genes (DEGs). The total number of DEGs in both groups peaked on Day 1 post-MVA vaccination, and by Day 2, only a small number of DEGs was observed. Therefore, we focused our analyses on Day 1 DEGs. The number of DEGs that were upregulated at Day

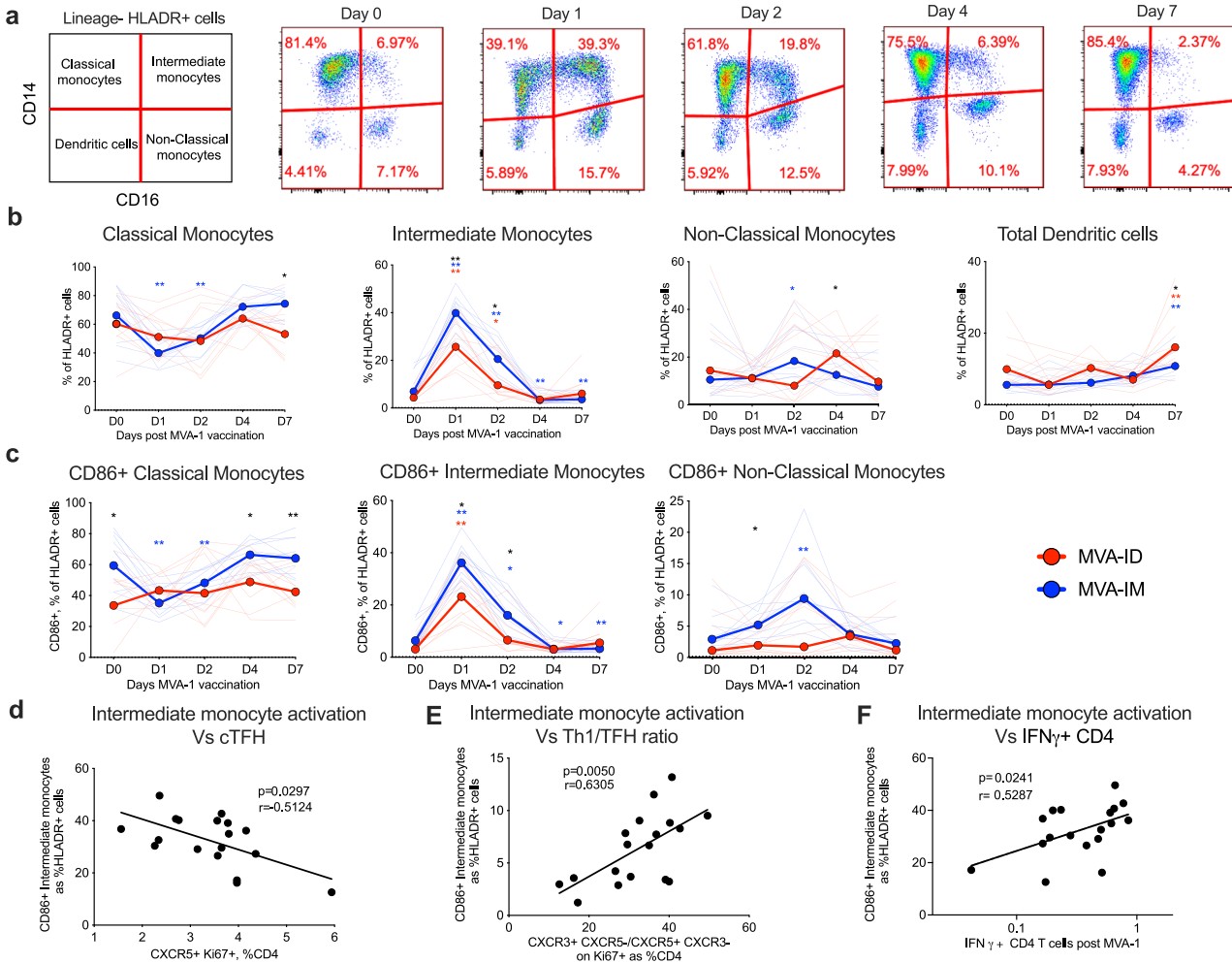

**Fig. 4 | Innate cell response post-MVA-1 immunization. a** Flow cytometry representative plots on lineage-HLADR+ cells indicating different monocyte subsets: classical monocytes (CD14 + CD16−); intermediate monocytes (CD14 + CD16+); non-classical monocytes (CD14− CD16+); dendritic cells (CD14−CD16−) at indicated time points post-MVA-1 immunization. **b, c** Frequencies of monocyte subsets, dendritic cells. **b** (classical monocytes: D7:black*$p$ = 0.015, D1:blue**$p$ = 0.003, D2:blue**$p$ = 0.009; intermediate monocytes: D1:black**$p$ = 0.007, D2 black*$p$ = 0.05, D1:red*$p$ = 0.003, D2: red*$p$ = 0.031, D1:blue**$p$ = 0.002, D2:blue**$p$ = 0.009, D4:blue**$p$ = 0.009, D7:blue**$p$ = 0.003; non-classical monocytes: D4:black*$p$ = 0.037, D2:blue*$p$ = 0.048; total dendritic cells D7:black*$p$ = 0.028, D7:red**$p$ = 0.003, D7:blue**$p$ = 0.002) and frequencies of activated (CD86+) monocyte subsets **c** (classical monocytes: D0:black*$p$ = 0.045, D4:black*$p$ = 0.014, D7:black**$p$ = 0.002, D1:blue**$p$ = 0.003, D2:blue**$p$ = 0.009; intermediate monocytes: D1:black*$p$ = 0.022, D2 black*$p$ = 0.017, D1:red**$p$ = 0.003,

D1:blue**$p$ = 0.002, D2:blue*$p$ = 0.013, D4:blue*$p$ = 0.013, D7:blue**$p$ = 0.003; non-classical monocytes: D1:black*$p$ = 0.017, D2:blue**$p$ = 0.002) at indicated time points post-MVA-1 immunization. Geomeans of all animals in respective groups were represented as bright lines and all individual animals were represented as faded lines. Groups were color-coded: MVA-ID − Red; MVA-IM − Blue. Asterisks were color-coded (MVA-ID − Red, MVA-IM − Blue) representing the vaccine groups and denote statistical significance in comparison to their respective Day 0 values, measured by Wilcoxon matched paired $t$ (two-tailed) test. Black asterisks in all the above data denote statistically significant differences between the groups at the given time point, measured by Mann–Whitney rank-sum test (two-tailed). **d**–**F** Spearman correlations (two-sided) between frequencies of Day 1 CD86+ activated intermediate monocytes and Day 7 **d** circulating Tfh, **E** Th1/Tfh ratio, **f** IFN-γ + CD4 T cells post-MVA-1 immunization. MVA-1: first MVA. Source data are provided as a Source Data file.

---

1 was markedly higher in the MVA-IM group (1123) compared to the MVA-ID group (814), and the majority (723) of these MVA-ID genes were also upregulated in the MVA-IM group. In contrast, a greater number of transcripts were downregulated in the MVA-ID group (327) compared to the MVA-IM group (235), with only 99 being common between the two groups (Fig. 5a). We performed a Reactome pathway enrichment analysis on these DEGs using networknalyst.ca[25]. We used a cutoff criteria of FDR <0.1 to enlist these enriched pathways. This analysis yielded pathways related to Interferon-alpha/beta signaling and cytokine signaling as the highly enriched pathways in both the groups but at relatively lower significance in the MVA-ID group than MVA-IM group (Fig. 5b). The interferon-alpha/beta signaling pathway consisted of some key genes like *IFI16*, a cytosolic DNA sensor; interferon receptors *IFNAR1* and *IFNAR2*; transcription factor *STAT-1* that is known to be associated with the type 1 interferon response with

cytosolic DNA stimulation[26,27]. This was consistent with MVA being a cytosolic DNA virus. Some pathways were significantly enriched only in the MVA-IM group. These included trafficking and processing of endosomal TLR, platelet degranulation, death receptor signaling, AIM2 inflammasome activation, and IL-1 signaling. In contrast, transcripts associated with the ISG15 anti-viral mechanism were enriched in the MVA-ID group (Fig. 5b.1). Among the downregulated transcripts, pathways associated with viral RNA and eukaryotic RNA translation and RNA metabolism were prominent but were enriched in both groups (Fig. 5c). While we observed a higher number of transcripts downregulated in the MVA-ID group compared to MVA-IM group, we could not find enrichment for specific pathways with these genes.

We next focused our efforts on foreign DNA-stimulated inflammasome response leading to proinflammatory cytokines since these pathways were more enriched in the MVA-IM group. We found striking

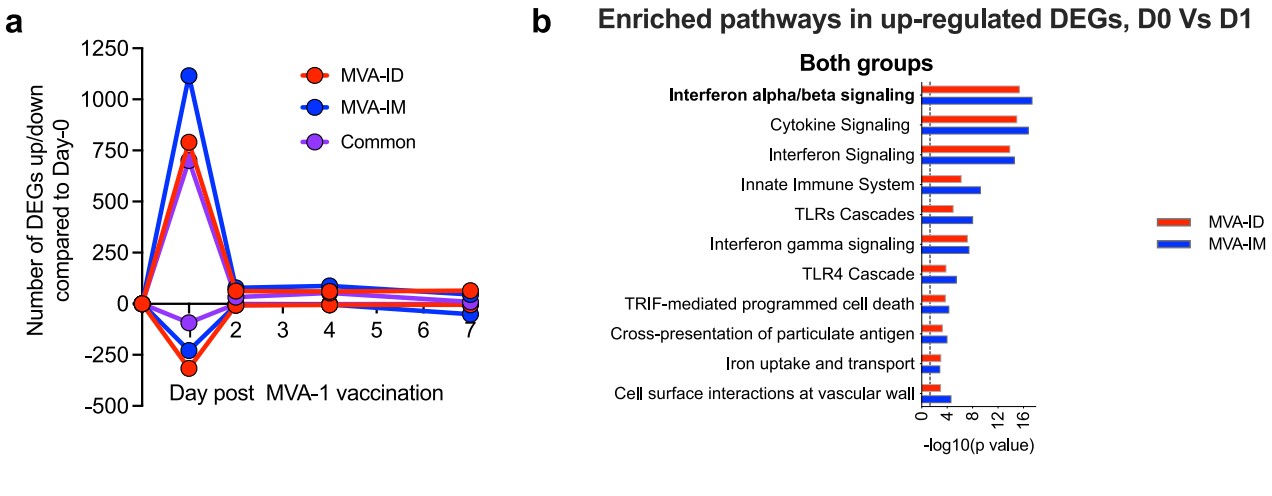

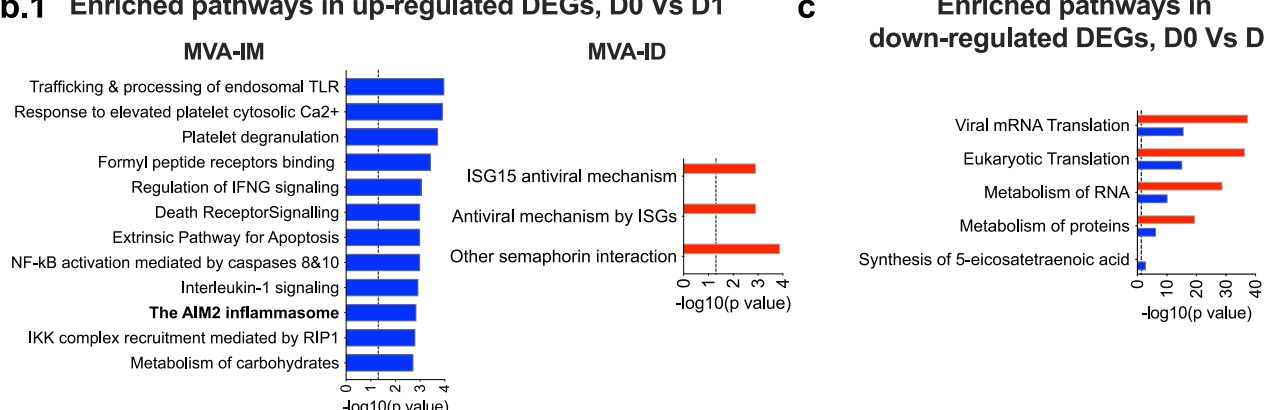

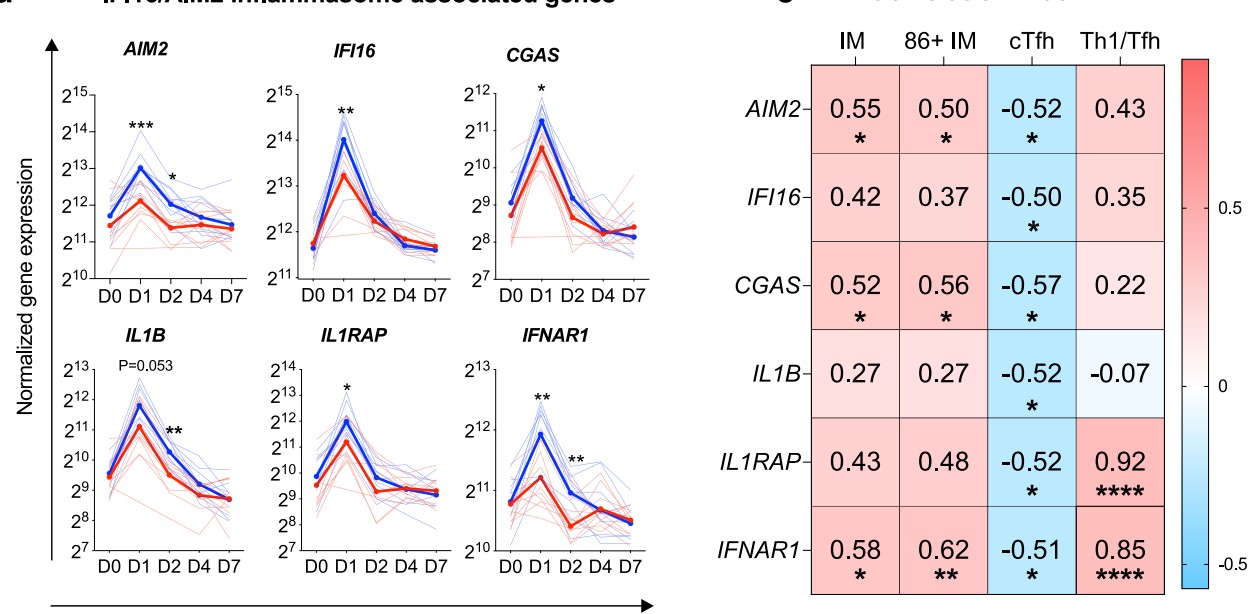

differences between the groups in the expression of transcripts for intracellular foreign DNA sensors like *AIM2*, *IFI16*, and *cGAS* genes that showed significantly higher induction in the MVA-IM group compared to the MVA-ID group. These genes encode proteins that bind to viral DNA and activate the inflammasome pathway. Consistent with higher *AIM2* transcripts, we also saw a high number of transcripts for *CASP1*

and *IL-1beta* in the MVA-IM group compared to MVA-ID group[28–30]. The expression of these genes peaked at Day 1 and returned to basal levels by Day 7 (Fig. 5d). These observations suggested that MVA-IM animals induced a higher proinflammatory response than MVA-ID animals leading to differential quality of adaptive immune response between the groups. Further validating this phenomenon, we observed a

**Fig. 5 | RNA sequence analysis in blood, post-MVA-1 immunization. a** Line graphs showing the number of DEGs in both the groups at indicated time points, when compared to their expression on Day 0 of MVA-1 immunization. The cutoff criteria was defined as $\log_2$fold change $\geq 1$ or $\leq -1$ and $p$ value $\leq 0.05$. A multiple-test correction was performed with the Benjamini–Hochberg method and a false-discovery rate <0.05. The lines were color-coded, representing groups (Red: MVA-ID, Blue: MVA-IM, Purple: number of genes that are up/down-regulated in both the groups). **b, c** Enriched reactome pathways associated with DEGs at Day 1, post-MVA-1 immunization, using online tool network analyst (networkanalyst.ca)[25]. The cutoff criteria used was $p$ value <0.05 and FDR <0.1. $P$ values are not adjusted for multiple comparisons. Bars were color-coded representing respective vaccine groups (MVA-ID – Red, MVA-IM – Blue), **b** pathway enrichment from upregulated DEGs, and **c** pathway enrichment from downregulated DEGs. **d** Longitudinal normalized gene expression of selected key genes along AIM2/IFI16 inflammasome pathway at Day 0, 1, 2, 4, and 7 post-MVA-1 immunizations. The bright color-coded lines represent the geometrical mean of the corresponding groups and light color-coded lines in the background represent individual animals in the corresponding groups.

Asterisks denote statistically significant differences between the groups at a given time point, measured by Mann–Whitney rank-sum (two-tailed) test (*AIM2*: D1:***$p$ = 0.0004, D2:*$p$ = 0.02; *IFI16*: D1:**$p$ = 0.003; CGAS: D1:*$p$ = 0.01; IL1B: D2: **$p$ = 0.0021; *IL1RAP*: D1:*$p$ = 0.035; *IFNAR1*: D1:**$p$ = 0.0057, D2:**$p$ = 0.0044). **e** Correlation matrix with genes along AIM2/IFI16 inflammasome pathway as variables on the y-axis and circulating Tfh, Th1/Tfh ratio, frequencies of intermediate monocytes (IM), and frequencies of activated intermediate monocytes (86 + IM) as variables on the x-axis. The positive correlation between the variables was denoted by a gradient of red color and the negative correlation between the variables was denoted by a gradient of blue color. In all the correlations included in the matrix, r and p values were included in the respective matrix box. Asterisks denote statistical significance from a spearman correlation (two-sided)(AIM2 Vs IM *$p$ = 0.017, 86 + IM*$p$ = 0.033, cTfh*$p$ = 0.027; IFI16 Vs cTfh*$p$ = 0.035; cGAS Vs IM *$p$ = 0.025, 86 + IM*$p$ = 0.015, cTfh*$p$ = 0.013; IL1B Vs cTfh *$p$ = 0.026; IL1RAP Vs cTfh*$p$ = 0.027, Th1/Tfh****$p$ < 0.0001; IFNAR1 Vs IM*$p$ = 0.011, 86 + IM**$p$ = 0.006, cTfh*$p$ = 0.031, Th1/Tfh****$p$ < 0.0001). MVA-1: first MVA. Source data are provided as a Source Data file.

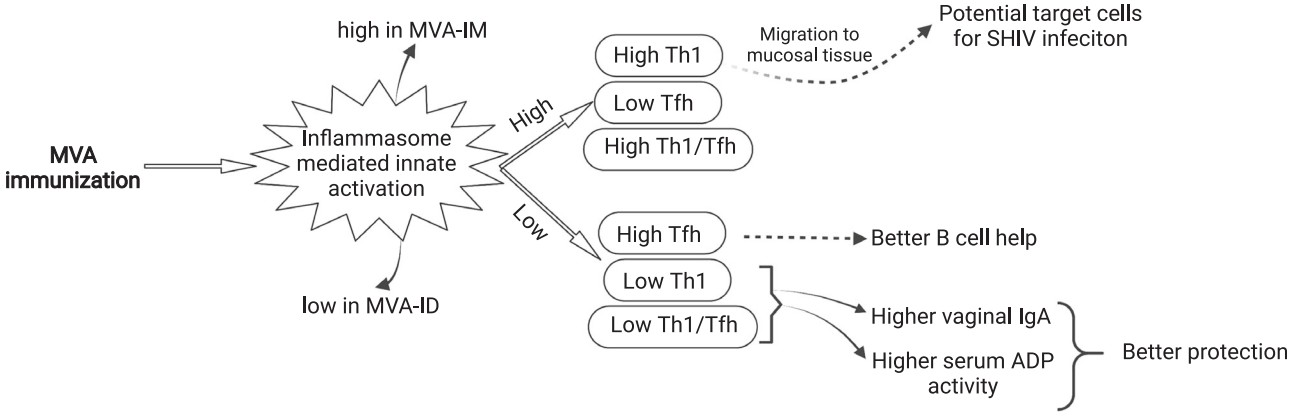

**Fig. 6 | MVA-induced innate response and its effect on shaping the adaptive immune response.** MVA immunization-induced inflammasome-mediated innate activation was higher in the MVA-IM group and lower in the MVA-ID group. The higher innate activation leads to high Th1, low Tfh and a high Th1/Tfh ratio. The migration of these higher numbers of Th1 cells to mucosal tissues could potentially lead to more number of target cells for SHIV infection. Lower innate activation leads to high Tfh, low Th1 and low Th1/Tfh. These higher Tfh cells provide greater B cell help. Low Th1 and low Th1/Tfh were associated with higher vaginal IgA and higher serum ADP activity, which, in turn, were associated with better protection. Solid arrows represent observed associations and dotted arrows represent predicted associations.

negative correlation between the expression of these AIM2/IFI16 inflammasome-associated genes and MVA-induced circulating Tfh response. These genes also correlated positively with the frequency of activated intermediate monocytes and higher Th1/Tfh ratio (Fig. 5e), suggesting a strong interplay between inflammasome response, intermediate monocytes, and circulating Tfh response. On a whole, our transcriptomic analysis of blood post-MVA immunization revealed that the MVA-IM group induced higher AIM2/IFI16 mediated inflammasome response than the MVA-ID group leading to differential quality of adaptive immune response and possibly thereby differential outcome of vaccine efficacy.

In summary (Fig. 6), our results showed that the MVA-IM and MVA-ID vaccinations induce different levels of inflammasome-mediated innate activation that was associated with significant changes in adaptive immunity. The higher inflammatory innate response was associated with the induction of higher Th1 cells, lower Tfh response, and higher Th1/Tfh ratio. The higher Th1 response, in turn, was associated with induction of lower vaginal IgA and serum ADP activities, the two variables that showed a direct association with protection in addition to a neutralizing antibody response. In addition, while high Tfh response could offer potential B-cell help, the higher Th1 response in blood could be detrimental to HIV vaccine efficacy as these cells could form potential target cells for infection in the mucosa, as reported in our earlier study[24]. The protected animals in the MVA-ID group had this unique combination of lower antigen-specific Th1 cells,

higher vaginal IgA, and higher ADP activity than the infected animals in the MVA-IM and MVA-ID groups (SFig. 8). Most important of these parameters could be the difference in Th1 response between the groups. This could have influenced the protective outcome between the vaccinated groups even though they had only marginal differences in other protection correlates of vaginal IgA and serum ADP activity. These results establish a link between the route of immunization, innate immune activation, adaptive immunity, and protection outcome against SHIV infection.

## Discussion

In the current study, we compared the effects of the route of MVA vaccination on the immunogenicity and efficacy of clinically translatable DNA/MVA/Protein vaccine approach using intravaginal SHIV challenges in rhesus macaques. We hypothesized that the route of vaccination could alter the magnitude and functional quality of the cellular and humoral immune response owing to the differences in the type and density of innate cells that are present in the skin and muscle[19–21]. Our results demonstrated that the DNA/MVA/protein vaccine protects rhesus macaques against the viral acquisition of homologous tier2 SHIV BG505 challenges but only when MVA was delivered via the ID route and not via the IM route. The MVA-ID group also showed protection or delayed acquisition even in animals with low-neutralizing antibody titers, which was not evident in the MVA-IM group. Surprisingly, there were no differences between the groups in

terms of the magnitude of the vaccine-induced CD8 T cell response or in antibody responses, including those with neutralization and other effector functions. Our results also demonstrated that while both routes of MVA vaccination showed control of viral replication in productively infected animals, the IM vaccinations were superior compared to ID vaccinations. Overall, their results highlighted the critical role of the route of MVA vaccination in determining protection from the acquisition of infection and viral control.

Our study demonstrated that the route of immunization significantly altered the T helper differentiation profile of vaccine-induced T cells. We observed a greater Tfh response, including circulating Tfh and GC-Tfh, and GC-B-cell responses post-MVA immunization in the MVA-ID group than in the MVA-IM group. We hypothesize that this reflects a better lymphatic drainage system from the skin than muscle tissue, where lymphatic capillaries are confined to fascial planes and not present in muscle bundles[31]. Similar observations of better lymphatic trafficking of antigens following vaccination via the ID route have been reported in mouse studies[32,33]. It should also be noted that we analyzed only inguinal LNs near the site of vaccination. Recent findings suggested that drainage of antigen to inguinal LNs could be variable following IM immunization, while iliac lymph nodes showed better drainage[34]. Generally, the antigen is drained more to LNs that are closest to the site of vaccination, and thus IM immunizations may target deeper LNs compared to ID immunizations. Since we sampled superficial LNs, it is possible that we did not sample the most optimal draining LNs for the MVA-IM group. However, systemic response in the blood post 1st MVA immunization also suggested higher Tfh response in the MVA-ID group than the MVA-IM group. This higher Tfh response in the MVA-ID group could also be an outcome of prior DNA immunization through the same ID route, whereas the other group received DNA and MVA through different routes, ID and IM, respectively. Though Tfh response was higher in the MVA-ID group, this did not reflect in the antibody response at Wk2 post-first MVA immunization, where the MVA-IM group showed a higher antibody response than the MVA-ID group. This could be due to higher levels of short-lived extrafollicular response in the LNs of the MVA-IM group, which contributes significantly to the antibody response at Wk2 post-first MVA immunization. Unfortunately, we did not measure antibody-secreting cells in the LNs.

In our recent study[24], we showed that animals that induce a stronger IFNg+ CD4 T cell response after MVA vaccination are likely to show poorer protection from intravaginal challenge. This was associated with higher migration of CCR5 + CD4 T cells to the vaginal mucosa. Consistent with these findings, we observed a negative influence of MVA-induced Th1 response on protection. Interestingly, protected animals in the MVA-ID group had lower IFNg+ CD4 T cells but higher SOSIP-specific vaginal IgA levels and serum ADP activity than infected animals in both groups. These data suggest that a combination of low IFNg+ CD4 T cell response with higher IgA and higher serum ADP activity was associated with the protection. Given that serum ADP activity and vaginal IgA were not significantly different between the groups, activated target cells at the vaginal mucosa of these animals could have played a potential role in modulating differential protection between the groups. Low-neutralizing animals of the MVA-IM group showed a trend of faster acquisition than even the control animals and further strengthens the potential role of activated target cells in vaginal mucosa. Analyzing the vaginal mucosa prior to the challenge would have given us a clearer picture of the key factors responsible for the difference in protection between the groups. We have not sampled vaginal biopsies during this time to avoid damage or inflammation at the site of the viral challenge.

Analysis of innate activation in blood post-vaccination revealed that the immunization route had significantly altered the activation of monocytes. To better understand these differences, we analyzed innate responses after the first MVA immunization in both groups and found higher frequencies of activated monocytes in the MVA-IM group than in the MVA-ID group. This difference was more striking in terms of frequencies of total and activated intermediate monocytes, which expanded on Day 1 post-vaccination in both the groups but to higher levels in the MVA-IM group than the MVA-ID group. These intermediate monocytes are known to be proinflammatory in nature and secrete IL−1b and TNF-a upon TLR stimulation[35–37]. Previous studies have shown that strong inflammatory cytokines like TNF-a and IFNg appeared to be blocking Tfh differentiation in malaria-like infections[38]. We saw a strong negative correlation between the MVA-induced frequency of intermediate monocytes on Day 1 and the circulating Tfh response on Day 7, suggesting an interplay between these intermediate monocytes and Tfh differentiation. Transcriptomic analysis after the first MVA immunization revealed a higher IFI16/AIM2-mediated inflammasome response in the MVA-IM group than in the MVA-ID group. Many key genes associated with this pathway were upregulated on Day 1 post-immunization and gradually reverted to basal level by Day 7. This could vary if the analysis was done much earlier, i.e., 6 h post-MVA immunization, as recent work done by ref. 39 showed the activation of innate immune pathways peaking at 6hr in MVA-ID immunized animals, while this activation of innate response continued to increase up to 24 h in MVA-IM immunized animals. However, globally Rosenabum et al., found the magnitude of this innate response to be less through the ID route of immunization. More interestingly, the genes associated with AIM2-mediated inflammasome response were negatively associated with circulating Tfh response and positively associated with frequencies of intermediate monocytes. Together, these observations provide key mechanistic clues to the differential immune response observed for the MVA-ID and MVA-IM groups post-MVA immunization. They strongly suggest the involvement of intermediate monocytes and inflammasome responses in regulating Tfh differentiation and thereby differentially shaping the adaptive immune response in the two groups.

Our study showed that the addition of protein boosts to DNA/MVA vaccination help to induce an autologous neutralizing antibody response. Two protein boosts with potent adjuvants like NP-3M-052 seem to be sufficient since the third protein boost did not significantly improve the titer. This adjuvant was earlier shown to induce high-magnitude antibody response and bone marrow resident long-lived plasma cells against HIV-Env[40]. The DNA/MVA vaccinations alone, despite encoding the VLPs with full-length and C-terminally truncated Envs, failed to generate a neutralizing antibody response that was above the detection limit of 1:20. This could be due to a relatively low titer of binding antibody response induced by DNA/MVA vaccination compared to the titer observed following protein boost. The neutralizing antibody titers that we observed with DNA/MVA/Protein vaccine regimen in this study were not higher compared to the titer generated by protein-only immunizations in our recent study[5]. Originally, we hypothesized that adding a prime immunization with membrane-bound trimeric protein would generate a higher neutralizing antibody titer after the SOSIP-protein boost. However, we did not observe this. Nevertheless, it is of note that we observed protection in the MVA-ID group even with low (1:50) or no (1:20) neutralization titers that we did not observe in our protein-only group[5], suggesting the contribution of immune parameters other than neutralizing antibody towards protection. After the final vaccination, there were no significant differences in both groups in terms of vaccine-specific systemic antibody response and IFNg+ CD4 T cells. This could be due to the strong booster immunizations with protein+3M-052-NP, which could have nullified the quantitative difference (if any) between the groups caused due to different routes of MVA priming. However, T cell responses like IFNg+ CD4 T cells could be much different in the vaginal mucosa, which we did not sample before the challenge in this study.

In conclusion, our data demonstrated that DNA/MVA/Protein vaccine could protect against the homologous tier2 intravaginal SHIV

BG505 challenges even in the absence of strong neutralizing antibody response and provide evidence that the route of MVA vaccination can significantly influence HIV vaccine efficacy by modulating innate cell activation and Th differentiation. One other alternative route of MVA vaccination worth exploring is subcutaneous immunization. This route of MVA administration in cynomolgus macaques showed better lymphatic draining of the antigen to the lymph nodes, lower Th1 response[39], and also induced higher activation of neutrophils long after immunization[41]. Our results strongly endorse the need for testing the influence of the route of immunization for HIV vaccines in humans.

## Methods

### Animals
Thirty-five Indian-origin female Rhesus Macaques were assigned for the study and maintained following NIH guidelines. The Institutional Animal Care and Use Committee (IACUC) of Emory University approved all the procedures involving animals. Animals were randomly distributed between the groups based on age and weight, and are detailed in STable 1. The study was conducted in parallel to the study described by us previously[5] in two large cohorts and the controls were shared between the two studies. See STable 1 for details.

### Immunogens
The DNA construct SHIV BG505-IRIS-CD40L (SFig. 1a) was designed from a base plasmid, pGA1/SHIV1086C_IRES CD40L, used earlier in our lab[15]. 1086C HIV-Env, Tat/Rev sequences from the base plasmid was replaced with BG505 HIV-Env and clade D derived Tat/Rev sequences using EcoR1 and Nhe1 restriction sites. REKR at the gp120-gp41 cleavage site was mutated to R6 to enhance furin-mediated cleavage. SIVmac239 *Gag* and *PRT* genes were mutated to inactivate RNA packaging into VLPs. Thirty-three amino acids at the carboxy-terminal of gp41 were replaced with amino acids from the SIVmac766 gp41 carboxy-terminal to match the sequence in the BG505 SHIV[42]. The plasmid was sequenced to make sure that no random mutations were incorporated during the process of cloning. This DNA was transfected into HEK-293T(ATCC, CRL-11268) cells using lipofectamine, and 48 h after transfection, the expression of recombinant proteins was confirmed by western blot and flow-cytometry; SIV Gag stained by 2F12 antibody (NIH AIDS reagent program), HIV-Env stained by PGT121 antibody for flow-cytometry staining, while anti-BG505 rhesus serum was used for western blot detection, CD40L stained by AF617 (R&D systems) (SFig. 1b, c). VLP formation was also confirmed by transmission electron microscopy (SFig. 1d).

rMVA-SHIVBG505_gp147 was constructed using a chimeric BG505 and clade D Env sequence (Gen Bank accession: KU958484.1 (https://www.ncbi.nlm.nih.gov/nuccore/KU958484.1/)) and SIV239 Gag, pol sequence. The envelope sequence was truncated at amino acid 738 to create gp147. We introduced additional E46K-A316W-T332N-A433P SOSIP (A501C, T605C, I559P) mutations into the Env sequence. In addition, we replaced amino acids E and K at positions 506 and 507 with RRRR to create an enhanced furin cleavage site and was optimized for MVA-preferred codons. This sequence was synthesized from GeneArt and was cloned using the Xmal site in pLW-73 with an independent mH5 promoter. It was subsequently recombined into MVA essential region expressing SIV Gag and Pol (at DeIII) (provided by B.Moss) between genes *18R* and *G1L* (SFig. 2a)[43]. rMVA was sorted using GFP (green fluorescent protein) and PGT121 antibody against surface Env, during the first round of selection. After seven rounds, plaques with GFP-negative rMVA/SHIV were picked, and the DNA sequences were confirmed. Gag and Env expression in rMVA/SHIV-infected cells were confirmed by western blotting (Gag: 2F12 antibody, Env: anti-BG505 rhesus serum) and flow cytometry (Gag: 2F12 antibody, Env: PGT121 antibody (SFig. 2b, c). VLP formation was observed through electron microscopy (SFig. 2d). Viral stock for immunizations was purified from rMVA/SHIV-infected DF1 cell (ATCC, CRL-3586) lysates using 36% sucrose cushion.

The protein immunogen, BG505 SOSIP.664 gp140 Env trimer, was obtained from the International AIDS Vaccine Initiative[5]. This recombinant Env trimer was produced in CHO cells(ATCC, CCL-61) using good manufacturing practice conditions[44].

### Immunizations
About 760 µg of DNA, SHIV BG505-IRIS-CD40L in 400 µl PBS was split into two doses and delivered ID followed by electroporation (BTX) into two thigh regions of all the vaccinated animals. About $1 \times 10^8$ PFU of rMVA-SHIVBG505_gp147 in 400 µl PBS was split into two doses and delivered ID into two thigh regions of the animal in the MVA-ID group. About $1 \times 10^8$ PFU of rMVA-SHIVBG505_gp147 in 1 mL PBS was split into two doses and delivered IM into two thigh regions of the animal in the MVA-IM group. About 200 µg of BG505 SOSIP.664 trimer protein with 75 µg of 3M-052 adjuvant (PLGA nanoparticles) in 1 mL of PBS was split into two doses and delivered subcutaneously into two thigh regions of all the vaccinated animals.

### SHIV virus challenge and quantification of viral load
All animals were subjected to weekly challenges with SHIV BG505.332 N.375YΔCT virus via the intravaginal route. The challenge virus had *Env* gene from BG505 HIV-1 virus and *Gag* gene from SIVmac766 virus. The *Env* gene was mutated to S375Y for enhanced binding and replication in rhesus CD4 T cells[42]. The virus stock was used at 1:3 dilution in a final volume of 1 mL serum-free RPMI-1640 medium, which amounted to 50 ng p27 or $3.5 \times 10^8$ virions per challenge. Animals that showed plasma viral load >60 copies/mL for 2 consecutive weeks, were considered infected and not challenged further. "Two weeks before the first detection of viremia" was defined as the "number of challenges for productive infection" in the corresponding animal. Viral load copy number was measured using quantitative real-time PCR for SIV *gag* gene[6]. All samples were measured in duplicates and mean values were reported. The vaccine efficacy was calculated based on the method by Hudgens and Gilbert[45]. Briefly, it was defined as the relative reduction in the per-challenge probability of transmission of the vaccinated compared to the un-vaccinated controls, assuming the chance of infection is independent between challenges.

### Measurement of binding IgG antibodies in serum
BG505 SOSIP gp140 specific binding antibodies in the serum were measured by enzyme-linked immunosorbent assay (ELISA)[15]. Briefly, assay plates were coated with 2 ug/mL antigen in PBS and incubated overnight at 4 °C. Known concentrations of serially diluted purified rhesus IgG (NHP reagent resource) were used as standards. The next day, these plates were washed, blocked, and then incubated with threefold serum dilutions for 1 hour. Standard IgG was captured using serially diluted anti-monkey IgG (Rockland). IgG bound to the antigen was detected using tetramethylbenzidine substrate (KPL, Gaithersburg, MD) and peroxidase-conjugated anti-monkey IgG (Accurate Chemical and Scientific, Westbury, NY). About 100 µl of 3N $H_3PO_4$ was added to stop the reaction. The concentration of antigen-bound IgG was estimated relative to the standard curve of known concentrations of purified rhesus IgG.

### Measurement of SOSIP-specific mucosal antibodies and serum IgA
Mucosal secretions were collected with Weck-Cel sponges[46] and the concentration of anti-SOSIP IgA and IgG in the secretions was measured using a binding antibody multiplex assay utilizing BG505 SOSIP.664 protein conjugated to BioRad BioPlexPro carboxylated magnetic beads as described in refs. 47,48. Serum IgA was also

measured using the same assay. Before performing IgA assays, samples were depleted of IgG as described[46] using Protien G Sepharose (GE Healthcare). Briefly, dilutions of standard or sample were mixed with protein-bound beads overnight at 4 °C. Beads were then washed and developed with biotinylated goat anti-monkey IgG or IgA (Rockland Immunochemicals) and neutralite avidin-phycoerythrin (Southern Biotech). Anti-SOSIP antibody concentrations in the samples were normalized by dividing by the total IgG or IgA concentration, measured by ELISA using plates coated with goat anti-monkey IgG or IgA (Alpha Diagnostics) and the above-biotinylated antibodies. The recombinant anti-HIV gp120 CD4 binding site rhesus b12 dimeric IgA antibody (NHP Reagent Resource) was used as a standard for both anti-SOSIP and total IgA assays. IgG purified from SHIV-infected macaques and pooled naïve macaque serum were used as standards for anti-SOSIP and total IgG assays, respectively.

### Measurement of neutralization titers in serum
Autologous neutralizing antibody $ID_{50}$ titers in the serum were measured using BG505.T332N Env pseudovirus[5,49,50]. Briefly, the pseudovirus was mixed with serially diluted heat-inactivated serum and added to TZM-bl cells (NIH AIDS reagent program, Cat. No. 8129). Forty-eight hours after infection, the cells were lysed and luciferase activity was measured by BioTek Cytation 3 or Synergy Neo2S multimode reader. Average background luminescence from uninfected wells was subtracted from each experimental well. Each experimental well was compared with a virus-only well without test reagent to represent 100% infectivity. All assays were performed with duplicate wells and repeated independently at least once. $ID_{50}$ titer values were calculated using a dose-response inhibition analysis function with variable slope, normalized y values, and log-transformed x values in GraphPad Prism-9.

### ADCVI assay
Cryopreserved human PBMCs were thawed and rested overnight at 37 °C, 5% $CO_2$ in a complete RPMI medium. The cells were then incubated with a heat-inactivated serum of vaccinated monkeys at 1:100 dilution in quadruplicate along with SHIV BG505-infected CCR5 + CEM-NK$^r$ cells. After 4 days of above incubation, cells were washed to remove any anti-Gag antibodies. The cells were then incubated for 3 more days and the virus released into the culture medium was quantified using p27 capture ELISA. Inhibition of viral infection was determined by dividing the p27 concentration in test wells by the average p27 concentration in wells that received pooled serum from adjuvant-only control animals[51]. The assay was repeated using PBMC from a different donor, and the results were averaged and represented as a median.

### ADCC assay
One million CCR5 + CEM-NK$^r$ cells encoding a tat-inducible luciferase promoter (kindly provided by Dr. David Evans, University of Wisconsin at Madison) were infected through spinoculation with 200 μL of the SHIV BG505.332N.375YΔCT virus for 3 h at 1500×g at 25 °C. Infection proceeded for 4 days. On Day 3 post-infection, SIV Gag protein levels were determined via flow cytometry with 2F12 antibody (NIH AIDS Reagent Program Cat# 1610). On Day4, infected targets were incubated with serum (1:100 dil) or monoclonal antibodies and rhesus CD16 expressing KHYG1 NK effector cells (from Dr. David Evans) at a 10:1 effector:target ratio for a period of 8 h. After incubation, 150 μl of the cell mixture was added to 50 μl of BriteLite Plus luciferase substrate reagent in a black 96-well plate (both from Perkin Elmer, Duluth, GA). Luciferase activity was measured after 2 min. ADCC activity was calculated as the percent reduction in luciferase when compared to effector and target cells alone[15,52].

### ADP assay
Antibodies in heat-inactivated day of challenge (wk70) serum were tested for their ability to enhance phagocytosis of SOSIP-labeled beads by THP-1 monocytic cells (ATCC, TIB-202)[47]. Briefly, avitagged BG505.664 SOSIP was biotinylated using a BirA biotin ligase kit (Avidity LLC, Aurora, CO) in accordance with the manufacturer's instructions. For each 96-well plate, $9 \times 10^7$ neutravidin-labeled FluoSpheres (Thermo Fisher, Waltham, MA) were mixed with 3.5 μg biotinylated SOSIP, washed, and incubated for 1 h with diluted serum. THP-1 cells ($2 \times 10^4$ per well) were then added. After a 4 h incubation in 5% $CO_2$ and 37 °C, the cells were washed, treated with trypsin for 5 min, washed, and fixed in 1% paraformaldehyde. The cells were then analyzed for fluorescence using a FACS Canto (BD Biosciences). Phagocytosis in each well was calculated by multiplying the % bead-positive cells by their median fluorescent intensity (MFI). After averaging the phagocytosis in triplicate cultures of test serum, this value was divided by the average phagocytosis for negative control macaque serum, at the same dilution, to obtain a final phagocytic score.

### Lymph node fine-needle aspirates (FNAs) and cell processing
Draining lymph nodes were identified by veterinarians and cells were collected using a 22-gauge needle attached to a 3 mL syringe by passing four times into the lymph node. These samples were suspended in RPMI medium containing 10% FBS, 1X penicillin/streptomycin. Samples were centrifuged, and ACK (ammonium-chloride-potassium) lysis buffer was added if the sample had red blood cells.

### Cellular phenotype analysis
Immune cell phenotypes were analyzed from blood, frozen PBMCs, and FNAs processed cells, as appropriately mentioned in the result section. Frozen PBMCs were carefully thawed, washed thrice with warm RPMI medium containing 10%FBS and 1X penicillin/streptomycin, and stained. Blood samples and frozen PBMCS were surface stained with respective innate cell marker antibodies and T cell phenotype antibodies (Innate panel: Apcy7 live/dead; A700 anti-human CD3(BD 557917, SP34-2, 3 μl in 100 μl); BV-650 anti-human CD20 (Biolegend, 302335, 2H7, 4 μl in 100 μl); BV-510 anti-human CD14 (Biolegend 301842, M5E2, 5 μl in 100 μl); PERCP anti-human HLADR (BD 347364, L243, 10 μl in 100 μl); APC anti-human CD66 (Miltenyi 130-118-539, TET2, 3 μl in 100 μl); BV-711 anti-human CD16 (BD 563127, 3G8, 5 μl in 100 μl); BV-605 anti-human CD86 (Biolegend 305430, IT2.2, 4 μl in 100 μl)) and incubated for 30 min at room temperature. After incubation, RBC in blood was lysed with 1X BD FACS lysing solution for 10 min. These cells were washed with FACS wash buffer and fixed with 1% PFA in PBS for 10 min and finally suspended in FACS wash buffer. FNAs processed cells and frozen PBMCs were surface stained with the following antibodies (FNAs panel: Apcy7 live/dead; PERCP anti-human CD3 (BD 552851, SP34-2, 5 μl in 100 μl); BV-650 anti-human CD4 (Biolegend 317436, OKT4, 0.1 μl in 100 μl); PEcy7 anti-human CD20 (Biolegend 302312, 2H7, 4 μl in 100 μl); BV-421 anti-human PD1 (Biolegend 329920, EH12.2H7, 3 μl in 100 μl); PE anti-human CXCR5 (Invitrogen 12-9185-42, MU5UBEE, 4 μl in 100 μl); APC anti-human CD8 (BD 340584, SK1, 3 μl in 100 μl), T cell phenotype panel: Apcy7 live/dead; A700 anti-human CD3 (BD 557917, SP34-2, 3 μl in 100 μl), BV-650 anti-human CD4 (Biolegend 317436, OKT4, 0.1 μl in 100 μl), BV-711 anti-human CD8 (Biolegend 344734, SK1, 5 μl in 100 μl), PE- anti-human CXCR5 (Invitrogen 12-9185-42, MU5UBEE, 4 μl in 100 μl), BV-605 anti-human CXCR3 (Biolegend 353728, G025H7, 1 μl in 100 μl) and incubated for 30 min at room temperature. These cells were washed with FACS wash and permeabilized with permeabilization buffer (Invitrogen cat 00-5123-43) for 30 min at room temperature. The cells were washed with wash buffer (Invitrogen cat 00-8333-56) and stained with the following intracellular marker antibodies (FNAs:FITC anti-human BCL6 (Biolegend 358514, 7D1, 5 μl in 100 μl); A700 anti-human Ki67 (BD 561277, B56, 5 μl in 100 μl), T cell phenotype: PEcy7 anti-human Ki67

(BD 561283, B56, 5 µl in 100 µl) and incubated for 30 min at room temperature. These cells were washed with wash buffer and finally suspended in FACS wash buffer. The cells were acquired with the Fortessa instrument (BD Immunocytometry systems, San Jose, CA) using BD FACS DIVA Software v8.0.1 and analyzed with FlowJo software (Treestar, Ashland, OR).

## T cell response

Antigen-specific T cell response was measured using the Intracellular Cytokine Staining (ICS) assay[22]. PBMCs were stimulated with over-lapping 1 µg/mL, BG505 Env peptide pools or 1 µg/mL, SIVmac239 Gag peptide pool(Cat: 12364, HIV reagent program) in the presence of co-stimulants anti-CD28 (BD 555725, CD28.2, 0.2 µl in 100 µl) and anti-CD49d (BD 555501, 9F10, 0.2 µl in 100 µl). BG505 Env pool (NIH ARP cat 13123) consisted of 213 overlapping 15mer peptides (overlapped by 11 residues). Env-1 pool consisted of the first 106 peptides and the rest was in Env-2 pool. After 2 h of stimulation at 37 °C in 5% $CO_2$, 0.5 µg/mL of Brefeldin-A (BD pharmingen) and 0.5 µg/mL of Golgistop (BD pharmingen) were added per each well. After 4 h of further incubation, cells were stored at 4 °C overnight. Next day morning, cells were washed in FACS wash buffer (PBS with 2% FBS and 0.05% sodium azide) and surface stained with PERCP anti-human CD3 (BD 552851, SP34-2, 5 µl in 100 µl), BV-650 anti-human CD4 (Biolegend 317436, OKT4, 0.1 µl in 100 µl), BV-510 anti-human CD8 (BD 563919, SK1, 0.02 µl in 100 µl) and live/Dead stain for 30 min at room temperature. Cells were washed and then permeabilized with cytofix/cytoperm (BD biosciences) for 20 min at 4 °C and washed with perm wash buffer (BD Biosciences). These cells were stained intracellularly with A700 anti-human IFN-γ (BD 557995, B27, 3 µl in 100 µl); PE-CF594 anti-human TNF-α (BD 562784, MAB11, 2 µl in 100 µl) and BV-421 anti-human MIP1β (BD 562900, D21-1351, 3 µl in 100 µl) for 30 min at 4 °C. Cells were washed with perm wash buffer and finally suspended in FACS wash buffer. These cells were acquired using the LSRII instrument (BD Immunocytometry systems, San Jose, CA) using BD FACS DIVA Software v8.0.1 and analyzed with FlowJo software (Treestar, Ashland, OR). The final values were determined by subtracting the background values in corresponding un-stimulated cells.

## RNA sequencing

Blood was collected at specified time points post-first MVA immunization into PAXgene Blood RNA tubes (BD Biosciences) and the RNA was extracted using the PAXgene Blood RNA Kit IVD (Qiagen). We avoided further stimulation of the cells in vitro to analyze transcriptomics ex vivo. RNA quality was assessed using an Agilent Bioanalyzer and then one microgram of total RNA was subjected to globin transcript depletion using the GLOBINclear Kit, human (Thermo Fisher Scientific). Ten nanograms of the globin-depleted RNA was used as input for cDNA synthesis using the Clontech SMART-Seq v4 Ultra Low Input RNA kit (Takara Bio), which uses oligo-dT priming for the reverse transcription step. This process would yield low recovery of non-coding RNAs, and hence we focused our analysis on protein-coding transcripts. Amplified cDNA was fragmented and appended with dual-indexed bar codes using the NexteraXT DNA Library Preparation kit (Illumina). Libraries were validated by capillary electrophoresis on an Agilent 4200 TapeSta-tion, pooled at equimolar concentrations, and sequenced on an Illumina HiSeq3000 at 100SR, yielding 25–30 million reads per sample. Reads were mapped to the MacaM version 7 assembly of the Indian rhesus macaque genomic ref. 53 available at: https://www.unmc.edu/rhesusgenechip/index.htm, using STAR version 2.5.2b with default alignment parameters[54]. One sample was excluded from the study, with <1% of reads uniquely mapping to the reference. Abundance estimation of raw read counts per transcript was done internally with STAR using the htseq-count algorithm[55]. DESeq2 version 1.22.1 R package[56] was used to produce normalized read

counts and compute the differential expression estimation using the Wald test. DESeq2 performs normalization for each gene in the dataset by fitting a generalized linear model. Read counts were modeled as a binomial distribution, then scaled by normalization factors, such as read depth, and also for gene-specific factors, such as length or GC content. Rather than showing housekeeping genes, we determine the efficiency of normalization on the dataset as a whole using relative log expression (RLE) plots (SFig. 9). A two-factor design using the vaccination group and time point was used. Within the group, differential expressions for post-vaccination time points were compared to pre-vaccination. Multiple-test correction was performed with the Benjamini–Hochberg method and a false-discovery rate <0.05 in addition to an absolute log fold change >1 was used to indicate statistical significance.

## Statistical analyses

The difference between the two groups was tested using a two-tailed non-parametric Mann–Whitney rank-sum test. For within-group comparisons, Wilcoxon matched-pairs test was used. For comparisons involving more than two groups, the Kruskal–Wallis test was performed prior to using the Mann–Whitney t-test between any two groups. All correlations were analyzed by the Spearman rank test. All Kaplan–Meier survival curves were analyzed using the log-rank test. GraphPad Prism v8 was used for all the analyses.

## Statistics and reproducibility

The expression of proteins by DNA and MVA vaccine constructs have been evaluated multiple times by western blotting and flow, as part of various other studies in the lab, and their expression was confirmed. TEM images were generated only once owing to their expensive nature, but we included appropriate controls while confirming the formation of VLPs.

We chose ten female rhesus macaques per vaccinated group and 15 animals per control group. We choose this number based on the power calculations as described earlier[57] and assuming an infection rate of ~0.3 in the control arm. These analyses revealed that we will achieve 80% or a higher power if the risk ratio (ratio of the rate of infection in vaccinated over the rate of infection in controls) is 0.37 or lower. In addition, NHP studies are expensive and there is a scarcity of animal availability, so we can't use group sizes much larger than this. Animals were randomly distributed between the groups based on age and weight. Veterinarians that performed NHP studies were blinded for study design. No data were excluded from the study.

Characterizations of immunogens, neutralization data, binding antibody data, and all in vitro experiments are reproducible. The viral challenge experiments are not repeated as it is a 2-year-long study, and it is expensive. In vitro experiments were repeated twice, and we have noticed similar observations. Analyses such as neutralizing antibody, viral load measurements, ADCVI, ADCC, and ADCP were done in a blinded fashion.

## Reporting summary

Further information on research design is available in the Nature Portfolio Reporting Summary linked to this article.

# Data availability

All data supporting the experimental findings of this study are available within the manuscript and are available from the corresponding author upon request. RNA-seq data used in this study is available in GEO repository under accession code GSE219118. Indian rhesus macaque genomic ref. 53 available at: https://www.unmc.edu/rhesusgenechip/index.htm. clade D Env sequence with Gen Bank accession:KU958484.1 is available at (https://www.ncbi.nlm.nih.gov/nuccore/KU958484.1/). Source data are provided with this paper.

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

## Acknowledgements

We thank all the veterinary staff of Yerkes National Primate Research Center, Emory University, for their support during this macaque study. We thank B. Wehrle and Z. S. Momin for their technical support in processing and storing of animal samples. We are grateful and thank J. P. Moore for useful discussions and providing us with BG505.SOSIP.664 protein for antibody binding assays. We thank A. Dey of the International AIDS Vaccine Initiative for providing us with BG505 SOSIP.664 for vaccinations. We thank CFAR Immunology/Emory Vaccine Center Flow Cytometry Core, especially K. Gill for the support with flow cytometry. We thank Bob Wilson for excellent technical assistance in the processing of secretions and measurement of mucosal antibodies. We acknowledge NHP Reagent Resource for Rhesus b12 dimeric IgA. We thank K. Gill, B. Cervasi, and CFAR Immunology/Emory Vaccine Center Flow Cytometry Core for their support with flow cytometry. We thank S. Wang and S. Liang of CFAR Immunology core for doing viral load assays. We thank W. Ding for the preparation of SHIV.BG505 challenge stock. We acknowledge NIH AIDS Reagent Program for SIV239 Gag and BG505 Env peptide pool reagents. This study was supported by NIH grants UM1 AI124436 (Emory Consortium for Innovative AIDS Research in Nonhuman Primates) to E.H. and R.R.A., the ORIP/NIH base grant P51 OD011132 to ENPRC, NIH grant P30AI050409 to Emory CFAR, and NIH grants AI26683 and OD010976 to Nonhuman Primate Reagent Resource.

## Author contributions

Conceptualization: R.R.A., E.H., and B.P. Funding: UM1AI14436, E.H. and R.R.A. Methodology: V.S.B., P.B.J.R., S.G., T.P.C., S.L.B., C.C.L., G.K.T., A.A.U., S.P.K., T.M.S., J.C.S., A.Y.S., T.L., A.S., and V.V. Investigation: R.R.A., E.H., B.P., V.S.B., C.A.D., D.M., P.A.K., S.E.B., G.M.S., M.T., and J.V. Funding acquisition: R.R.A. and E.H. Project administration: R.R.A. and V.S.B. Supervision: R.R.A. and E.H. Writing—original draft: R.R.A. and V.S.B. Writing—review and editing: R.R.A., V.S.B., and coauthors.

## Competing interests

Rama Amara is a co-inventor of the DNA/MVA vaccine technology that has been licensed to Geovax Inc., by Emory University. The remaining authors declare no competing interests.

## Additional information

[1]Emory Vaccine Center, Division of Microbiology and Immunology, Emory National Primate Research Center, Emory University, Atlanta, GA 30329, USA. [2]NHP Genomics Core Laboratory, Emory National Primate Research Center, Atlanta, GA 30329, USA. [3]Department of Surgery, Duke University School of Medicine, Durham, NC, USA. [4]Department of Microbiology, Immunology, and Parasitology, Louisiana State University Health Sciences Center, New Orleans, LA, USA. [5]Department of Pathology and Laboratory Medicine, Emory Vaccine Center, Emory National Primate Research Center, Atlanta, GA, USA. [6]Rollins School of Public Health, Emory University, Atlanta, GA 30322, USA. [7]3M Corporate Research and Materials Lab, Saint Paul, MN, USA. [8]3M Drug Delivery Systems, Saint Paul, MN, USA. [9]Department of Medicine, University of Pennsylvania, Philadelphia, PA, USA. [10]Department of Microbiology and Immunology, Stanford University School of Medicine, Stanford University, Stanford, CA, USA. [11]Department of Microbiology and Immunology, Emory School of Medicine, Emory University, Atlanta, GA 30322, USA. [12]Present address: Department of Laboratory Medicine and Pathology, University of Washington, Seattle, WA, USA. ✉e-mail: ramara@emory.edu

