## [Peer Review File · Nature Communications]

Intradermal but not intramuscular modified vaccinia Ankara immunizations protect against intravaginal tier2 simian-human immunodeficiency virus challenges in female macaquesREVIEWER COMMENTS

Reviewer #1 (Remarks to the Author):

This manuscript details exciting findings on how intradermal vaccination with MVA as part of a DNA/MVA/protein vaccine regimen provides better protection when compared to IM MVA in a intravaginal SHIV challenge model in rhesus macaques. Whereas both groups developed comparable levels of neutralizing and nonneutralizing antibodies only the ID vaccinated animals showed significant differences in acquisition. The authors attempted to tease out the mechanisms behind this by excluding animals with high neutralizing antibody levels and looking at animals exhibiting protection in the absence of these responses and found that protection correlated with vaginal IgA and serum ADP levels. The authors also found higher germinal center B cell and Tfh responses in peripheral lymph nodes and differences in the CD4 skewing towards Th1 or Tfh as well as the activation of intermediate monocytes and the inflammasome. The demonstration that route of vaccination plays a significant role in the types of responses and the degree of protection as well as looking at the mechanisms leading to these differences is an important contribution to the field.

There is a lot of support for the conclusions provided by the authors, including extensive evaluation of systemic and mucosal antibody responses, flow cytometry, transcriptomics, and assessment of viral acquisition. The vaginal mucosa was not assessed directly in this study, which limits direct evidence of cellular responses and barrier characteristics, however comparisons are made to prior work involving similar vaccinations that demonstrated significant changes in this compartment.

A number of the key assessments were performed 1 week after the first MVA vaccination based on prior work demonstrating that this was a key time point based on peak CD4 T cell responses. At this time, despite higher circulating proliferating Tfh and higher lymph node GC-Tfh and GC-B cell responses in the ID group the IM group actually had significantly higher levels of serum IgG and levels of vaginal antibodies were not different between groups. It would be important to address this potential contradiction that the levels of antibodies appear better in the IM group at a time when the levels of cellular populations that should contribute to/correlate with a better Ab response are better in the ID group. Additionally, after the final vaccination there were no significant differences in serum or vaginal antibodies, nor in levels of the key INF γ + CD4 T cells between groups. It appears that there may be different mechanisms of protection working in different animals within the ID group and these may be made less effective in the IM group based on increased in activated target cells in the vaginal mucosal seen in prior work. The lack of these samples in the current work leaves open the question of what happens in the ID animals.

It would be important for the authors to comment on the potential to have sampled the wrong lymph nodes for the IM group leading to some of the discrepancies noted above. Published work suggest that the drainage from the IM location to the inguinal nodes can be variable with the iliac nodes being a more consistent draining site for injections in the quadriceps (Barber-Axthelm IM, Kelly HG, Esterbauer R, Wragg KM, Gibbon AM, Lee WS, Wheatley AK, Kent SJ, Tan HX, Juno JA. Coformulation with Tattoo Ink for Immunological Assessment of Vaccine Immunogenicity in the Draining Lymph Node. *J Immunol.* 2021 Jul 15;207(2):735-744. doi: 10.4049/jimmunol.2001299. Epub 2021 Jul 9. PMID: 34244296.). My experience with IM injections in the arms is that they typically drain to deeper lymph nodes in the axillary region meaning that there is the potential for the differences in the LN responses to be that the sites sampled drained the ID and not the IM locations of injections given the needle sampling technique described would seem to be more likely to target the more superficial nodes. One additional point the authors should address is the potential for the prior DNA vaccine being given ID to influence the outcome of the ID vs IM MVA responses. I.e. if the ID MVA went to the same lymph nodes as the ID DNA is there the potential for a different

response vs the IM where the draining lymph nodes may be different from the site of DNA injection?

There appears to be a trend toward significance in terms of acquisition in ID animals with low neutralization titers vs others, however this is not significant and statements in the manuscript indicating a difference should make sure it is clear that this is not a statistically significant difference. It also appears that there is a trend for the IM low Neut animals to acquire faster than the controls, which would be consistent with the authors prior work indicating that these animals have more activated target cells in the vagina and the potential for these to increase acquisition risk. This potential for increased activated target cells to increase risk of acquisition is a bit of a confounding variable as it is unclear how this compares between the ID and IM groups as the vaginal mucosa was not assessed in the present study. In fact, as levels of vaginal IgA and ADP don't differ significantly between groups overall, it is possible that the increase in activated target cells in the IM group is a key reason why the low Neut animals acquired quickly and if these were not increased in the ID group it could be a significant factor leading to reduced acquisition, especially when combined with the demonstrated differences in IgA and ADP responses.

When comparing acquisition in the animals with high and low INF γ + CD4 T cells the route of vaccine must be excluded and both IM and ID animals meeting the definitions included in order to achieve a statistically significant result. This suggests that in addition to the route of vaccine there are other factors such as background inflammation at the time of vaccination or other animal intrinsic factors that must contribute to the generation of these responses and these factors should be evaluated if possible and discussed. In fact, in figure 4 it appears that the background inflammation may in fact be different between the two groups prior to receiving MVA (if D0 is collected prior to MVA vaccination) as the activated proinflammatory CD86+ classical monocytes are significantly higher (twice the level) in the IM group compared to the ID group at the baseline time point.

Why is there no basal data for the IM group on the GC-Tfh and GC-B cell graphs in fig 3?

The authors do a good job in Fig 6 of pointing out the potential for increased target cells as a potential factor in the differences in acquisition, but this may need to receive more attention in the text as a potential reason why small differences in levels of vaginal IgA and serum ADP, that are not significantly different between the IM/ID groups as a whole, lead to significant differences in acquisition based on the different routes.

Reviewer #2 (Remarks to the Author):

The manuscript "Intradermal but not intramuscular MVA immunizations protect against intravaginal tier2 SHIV challenges in macaques" by Bollimpelli et al. pertains to how routes of administration of the modified vaccinia Ankara elicited unique protection profiles in macaques. The authors hypothesize that a particular route of vaccination "... could alter the magnitude and functional quality of the cellular and humoral immune response owing to the differences in the type and density of innate cells present in the skin and the muscle". The authors observed enhanced protection in the intra-dermally immunized group compared to the intra-muscular group and conduct a series of immunological studies to probe the correlates of immunity. These studies pointed to enhanced mucosal IgA and serum phagocytic activity, linked to enhanced Tfh responses and reduced inflammatory intermediate macrophage signatures providing unique insights for next generation vaccine design. This study corroborates previously defined correlates of immunity observed across human and non-human primate studies and points to a new strategy to tune antibody quality that may be key to protection against HIV. There is a desperate need for enhanced

multivariate and even machine learning analyses to help explain the relationship between the immune correlates both run in a discriminant as well regression mode. There are a few addressable issues listed below:

Major

1. The statistical grouping of viral loads in Figure 1C is not clear. The asterisks' presumably indicate the significance of that group (MVA-ID and MVA-IM) from controls. But are there additional differences within groups? The comparisons can be clarified with lines on the figures.
2. The authors find an association between neutralization in the IM group and mucosal IgA and peripheral ADP with viral control in the ID-MVA group, however whether these features are correlated, or explain protection orthogonally is unclear. A multivariate model could provide critical insight on the independent predictive accuracy of each feature in protection. It would be interesting to see if the signature of protection can be enriched if both arms are combined, with a model accounting for different immunization strategies.
3. Along the same lines, the enhanced protection against infection in ID but potentially enhanced viral control observed with IM vaccination is quite interesting. Correlates of viral control are not actively investigated in this study. A similar multivariate model probing the correlates of viral control, with animal groups combined, could point to additional unique immune correlates of viral control.
4. For Figures 3A and 3B, the basal group representation as belonging to MVA-ID alone is confusing. The authors could greatly clarify these figures by removing the basal column and quantifying % changes or fold changes of MVA-ID and MVA-IM over basal.
 - a. a % or fold increase in immune cell phenotypes from basal could provide enhanced clarity on differences across groups.
5. The authors examine the differences in Th/Tfh and transcriptional profiles across the ID and IM arms but do not probe whether particular profiles or signatures explain IgA levels, ADP and neutralization.
6. Looking at the manufacturer's protocols for the Ultra Low Input RNA kit, it is unclear if this is a bulk reverse transcription or an oligo-dT-based protocol. If this was a bulk reverse transcription, are there differences between the groups in non-coding RNAs? If this was an oligo-dT-based reverse transcription, then the authors need to clearly state that in their methods.
7. The read counts standardizing for RNAseq need to be articulated more clearly. It would also help if the authors included a few housekeeping genes that remained constant between the two groups such as GAPDH, RPL19, etc.

Minor

1. The EM images in SFig1 and SFig 2 should be at the same magnification and require a scale bar.
2. Were some of the blots in SFig 1B run on the same gel? It looks like the lanes of the ladder do not correspond to the same gel, which is fine if they were run at the same time. But it should be disclosed.
3. There is no reference to panel A in the Figure 2 legend.
4. The black and blue dots in Figure 2E, H, and I, and Figure 3G-I are difficult to discern. Please consider using a lighter shade of blue, or using gray instead of black.
5. The legend font in Figures 3A, 3B, and 3E is extremely small and borderline non-legible
6. The titles of the graphs in SFig 4 are cutoff.
7. SFig 6B and SFig 6C need to be incorporated into the Main Figures. They are far too important to the authors' conclusions to be the final two panels in the supplemental figures.
8. Articles are missing throughout the manuscript and make it difficult to read.

Reviewer #3 (Remarks to the Author):

The authors previously demonstrated that 1) protein-only vaccination protects from autologous challenge in rhesus macaques and that protection is associated with neutralizing titer above 1:300 and 2) when combined with heterologous prime-boost with viral vector vaccines, some animals with neutralizing antibody (NAb) titers below 1 :300 are protected (Arunachalam et al, 2020). Here the authors report with another heterologous prime-boost regimen, that the route of administration of MVA-vectored vaccine matters for protection. When MVA is administered intradermally (MVA-ID), macaques are protected against the acquisition of infection, whereas they are not when MVA is injected intramuscularly (MVA-IM). However, MVA-IM-vaccinated animals exhibit a better control of viral replication. Interestingly, similar though heterogenous NAb titers are detected in both groups (ranging from below detection limit to 1:6,000) and in both groups, titers above 1:50 correlate with protection. In the MVA-ID group only, some animals with neutralizing titers below 1 :50 are protected and protection is associated with Ab-dependant phagocytic activities (ADP) in serum and with vaginal IgA binding titers. MVA-ID immunization is also associated with higher germinal center Tfh and B-cell responses, and in blood, with a lower ratio of Th1 to Tfh cell activation and a lower activation of intermediate monocytes and inflammasome. Overall, the data are of interest to the HIV vaccine field and more largely to the vaccine field, emphasizing the need of testing more in depth the impact of immunization route. Results are quite clearly presented and the manuscript is most of the time well written. However, language and style can be improved in some sections (see minor issues) and some revisions are needed for publication.

Major concerns

1. The title, the abstract and the end of the introduction (lines 96-97) emphasize the difference of protection between MVA-ID and MVA-IM groups. Actually, MVA-ID group is protected against the acquisition of infection and both group are protected against high viral loads in breakthrough infection, with MVA-IM doing even better than MVA-ID (Fig. 1). The authors should reformulate to align with the message of the conclusion section (line 374 to 377), which is clearer and more in line with the data.
2. The correlates of protection against high viral load should also be analyzed comprehensively (see below).
3. The Control group seems to be the same as in Arunachalam et al, 2020. This should be indicated in the Materials & Methods Animals section.
4. The authors should indicate if all animals were challenged simultaneously in the Materials & Methods Animals section. If not, the authors should precise to which cohort belong each animal in Sup Table 1.
5. The animal groups are quite heterogeneous in terms of age: Control group median 3 years range 12, MVA-IM group median 6 years range 4, MVA-ID group median 7.5 years range 9. The author should indicate whether there is a significant difference and if it has an impact on the results, particularly if the protection outcome is different according to the age.
6. As the ages are quite different, the authors should add the animal weights in Sup Table 1.
7. The authors should provide details on the randomisation method they used.
8. Line 100 : the authors should precise that the protection that is observed in MVA-ID and not in MVA-IM is against the acquisition of infection
9. Figure 1

- a. Title : the authors should precise that MVA-ID protects against the acquisition of infection
 - b. B. Though the Control group is the same as in Arunachalam et al, 2020, the survival curve is slightly different : the authors should correct or explain
 - c. B. The authors should provide calculation details for vaccine efficacy per challenge in the legend or in the Mat&Meth section
 - d. C. The authors should add a 5th graph, with statistical analysis, to compare the areas under the curve of viral RNA kinetics from week 0 to week 11 for the three groups
 - e. C. Kruskal-Wallis test should be performed when more than 2 groups are compared before comparing groups using Mann-Whitney test. They authors perform this test and mention it in the legend and in the Statistical analysis section accordingly.
 - f. C. The authors should check for consistency or clarify: in the legend, week 1 is defined as the first detection of positive viral load, which is fine, whereas my understanding from line 440-441 is that it is week 2.
10. The authors should provide a supplementary figure with the viral load curves of each animal, with animal IDs, according to the time of the first challenge. This will give insight on whether viral replication is different when animals are infected at the first challenges or later on.
11. The analysis of the correlates of protection is quite confusing.
- a. It is not clear enough whether all animal with neut > 50 were protected (lines 158-160)
 - b. The authors should not analyze the MVA-ID animals that got infected after 9 challenges along with the protected ones as the delay of infection can be stochastic especially since one animal in the Control group got infected after 9 challenges too.
 - c. Line 161 : the difference is stated as significant whereas it is not the case in Fig 2F.
 - d. As there is a delay of 10 weeks between the first and the last challenge, the Neut data at the time of infecting challenge for the breakthrough animals and at the end of the 10 challenges for the protected ones would be of high interest to explore 1) the durability of the response and 2) the impact of repeated vaginal exposures to vaccine-induced responses.
 - e. Figure 2E :
 - i. There should be only one data plotted above the 10th challenge as there is only one animal that got infected at this challenge ; the same for Fig. 2H and 2I
 - ii. The data of animals that remained protected from SHIV acquisition should not be plotted at all in this graph ; the same for Fig. 2H and 2I.
 - iii. The threshold line should be plotted at 50 and not at 20 as 50 is the value chosen by the authors to discriminate between low and high Neut. This should also be precised in the legend.
 - f. Figure 2F : the authors should add the curves of the high neut animals, for each group
 - g. Please analyze the correlates of protection with the same type of graph as Sup Fig 3H, but discriminating only by « infected » vs « protected », along with Mann-Whitney test, for all the parameters that were studied

h. Please provide the analysis of the correlates of protection against high viral load as in Sup. Fig. 5A, for all the parameters that were studied. It is only provided for binding BG505-specific Ab (line 176) and it would be of high interest for the other ones (neut, mucosal Ab titers etc)

i. For improved clarity, the Neut values at the time of the 1st challenge should be added in the Sup Table 1.

j. I would suggest to represent the correlates of protection and their associations by a correlation matrix heatmap with Ab and cellular parameters in order to have a global overview.

12. Lines 309-311 : the authors should precise that the protection they are talking about is against the acquisition of infection.

13. The results should be discussed in light of two recent publications that studied the impact of the route of MVA immunization in macaques (Rosenbaum et al, 2021, <https://doi.org/10.3389/fimmu.2021.645210> and Feraoun et al, 2022, <https://doi.org/10.3389/fimmu.2021.784813>).

Minor issues

14. Please clarify the role of CD40L in the DNA vaccine

15. References 5 and 6 are the same

16. Lines 151-153 : « On the day of first challenge » ; then check for the wording which seems odd or remove « ID50 is less than 1 :20 »

17. Line 159 : check for odd wording or remove « were »

18. Line 185-187 : please simplify the wording ; I would suggest « The frequency of germinal center T follicular helper cells (GC-Tfh) ranged from 1-4% (average 2%) at baseline and remained unchanged after the two DNA vaccinations »

19. Line 200 : than in the MVA-IM group

20. Fig. 2

a. Legend line 853: « A) » is missing

b. Graph D : title of the X « two weeks post immunization » is not convenient for the last data plotted which are the data of the day of challenge. Please correct by separating the X axe for instance

c. Graph H : typo in the title

21. Fig Sup. 3 : add the group color code in the figure for improved clarity

22. Typos

a. Check abbreviations: liter is L and not l

b. vs : italicize

c. Add a space between number and unit, i.e. line 424 « 760 µg » (many occurrences)

d. Line 425 and others: correct « 1x10⁸ PFU » (several occurrences)

Dear Reviewers,

We are so thankful and grateful for reading the manuscript thoroughly and for your highly encouraging comments about this work in general. We have taken all of your comments/concerns into consideration and revised the manuscript accordingly. Below, please find our point-by-point response to each comment/critique in **RED**. We hope that you will be satisfied with our revisions and response, and the revised manuscript will be suitable for publication.

Response to Reviewer #1:

This manuscript details exciting findings on how intradermal vaccination with MVA as part of a DNA/MVA/protein vaccine regimen provides better protection when compared to IM MVA in a intravaginal SHIV challenge model in rhesus macaques. Whereas both groups developed comparable levels of neutralizing and non-neutralizing antibodies only the ID vaccinated animals showed significant differences in acquisition. The authors attempted to tease out the mechanisms behind this by excluding animals with high neutralizing antibody levels and looking at animals exhibiting protection in the absence of these responses and found that protection correlated with vaginal IgA and serum ADP levels. The authors also found higher germinal center B cell and Tfh responses in peripheral lymph nodes and differences in the CD4 skewing towards Th1 or Tfh as well as the activation of intermediate monocytes and the inflammasome. The demonstration that route of vaccination plays a significant role in the types of responses and the degree of protection as well as looking at the mechanisms leading to these differences is ***an important contribution to the field.***

We are pleased to know that the Reviewer thinks our findings on the route of vaccination influencing the types of immune responses, the protection outcome and the associated mechanisms are an important contribution to the field. We are thankful to the Reviewer for these encouraging comments.

There is a lot of support for the conclusions provided by the authors, including extensive evaluation of systemic and mucosal antibody responses, flow cytometry, transcriptomics, and assessment of viral acquisition. The vaginal mucosa was not assessed directly in this study, which limits direct evidence of cellular responses and barrier characteristics, however comparisons are made to prior work involving similar vaccinations that demonstrated significant changes in this compartment.

1. A number of the key assessments were performed 1 week after the first MVA vaccination based on prior work demonstrating that this was a key time point based on peak CD4 T cell responses. At this time, despite higher circulating proliferating Tfh and higher lymph node GC-Tfh and GC-B cell responses in the ID group the IM group actually had significantly higher levels of serum IgG and levels of vaginal antibodies were not different between groups. It would be important to address this potential contradiction that the levels of antibodies appear better in the IM group at a time when the levels of cellular populations that should contribute to/correlate with a better Ab response are better in the ID group.

We agree with the reviewer's observation. In data not presented in the current manuscript, we observed a higher plasmablast response at day 4 post 1st MVA vaccination in the IM group than ID group. As peak antibody response at Wk2 post vaccination is mostly dominated by short-lived plasmablast response, the higher antibody response in the IM group post 1st MVA vaccination could be due to short lived extrafollicular antibody response. We have added this to the discussion (lines 363-369).

2. Additionally, after the final vaccination there were no significant differences in serum or vaginal antibodies, nor in levels of the key INF γ + CD4 T cells between groups.

We think strong booster immunizations with protein+ 3M-052-NP induced robust Ab responses in both groups and that could have nullified the quantitative differences (if any) between the groups caused due to MVA priming. Coming to IFN-g+ CD4 T cell response, it was measured at the systemic level but not at the site of viral challenge i.e vagina. Here, we showed that IM group had higher Th1/Tfh ratio post 1st MVA vaccination and in our earlier study we showed that these systemic Th1 responses can translate into target cells at the vaginal mucosal surface (Chamcha et al., Science Translational Medicine 11(519):eaav1800). We did not sample vaginal mucosa in this study to avoid causing any potential damage or inflammation at the challenge site. We believe that these target cells at mucosa could be one of the potential reasons for the differences in viral acquisitions between the groups. We have added this to the discussion (lines 428-434).

3. It appears that there may be different mechanisms of protection working in different animals within the ID group and these may be made less effective in the IM group based on increased in activated target cells in the vaginal mucosal seen in prior work. The lack of these samples in the current work leaves open the question of what happens in the ID animals.

We agree with the Reviewer's opinion on activated target cells in vaginal mucosa and their importance. However, as mentioned above we did not sample the mucosa to avoid any potential damage or inflammation at the challenge site as is normally practiced in SIV/SHIV vaccine protection studies. We acknowledged this limitation in the discussion (lines 382-385).

4. It would be important for the authors to comment on the potential to have sampled the wrong lymph nodes for the IM group leading to some of the discrepancies noted above. Published work suggest that the drainage from the IM location to the inguinal nodes can be variable with the iliac nodes being a more consistent draining site for injections in the quadriceps (Barber-Axthelm IM, Kelly HG, Esterbauer R, Wragg KM, Gibbon AM, Lee WS, Wheatley AK, Kent SJ, Tan HX, Juno JA. Coformulation with Tattoo Ink for Immunological Assessment of Vaccine Immunogenicity in the Draining Lymph Node. J Immunol. 2021 Jul 15;207(2):735-744. doi: 10.4049/jimmunol.2001299. Epub 2021 Jul 9. PMID: 34244296.). My experience with IM injections in the arms is that they typically drain to deeper lymph nodes in the axillary region meaning that there is the potential for the differences in the LN responses to be that the sites sampled drained the ID and not the IM locations of injections given the needle sampling technique described would seem to be more likely to target the more superficial nodes.

We agree with the Reviewer that based on the studies outlined by the Reviewer and emerging data from NHPs clearly show that antigen drainage to the lymph nodes varies significantly with the route of immunization, and this can influence the GC response. Because of this we could have potentially missed sampling the most optimal draining LN for both MVA-ID and MVA-IM groups, and more so for the latter. For this reason, we also studied circulating Tfh responses that can provide additional insights about responses in tissue soon after vaccination. Our data on systemic response in the blood post the 1st MVA immunization also suggested higher Tfh response in MVA-ID group than MA-IM group. We have clarified this in the discussion section and cited the reference suggested by the Reviewer (lines 353-361).

5. One additional point the authors should address is the potential for the prior DNA vaccine being given ID to influence the outcome of the ID vs IM MVA responses. I.e. if the ID MVA went to the same lymph nodes as the ID DNA is there the potential for a different response vs the IM where the draining lymph nodes may be different from the site of DNA injection?.

This is a possibility and we acknowledged this in the discussion (lines 361-363).

6. There appears to be a trend toward significance in terms of acquisition in ID animals with low neutralization titers vs others, however this is not significant and statements in the manuscript indicating a difference should make sure it is clear that this is not a statistically significant difference.

The difference in the rate of infection acquisition between MVA-ID and MVA-IM animals is significant, and it was a trend between MVA-ID and Controls. We modified the results section accordingly (lines 178-179, 183-185).

7. It also appears that there is a trend for the IM low Neut animals to acquire faster than the controls, which would be consistent with the authors prior work indicating that these animals have more activated target cells in the vagina and the potential for these to increase acquisition risk. This potential for increased activated target cells to increase risk of acquisition is a bit of a confounding variable as it is unclear how this compares between the ID and IM groups as the vaginal mucosa was not assessed in the present study. In fact, as levels of vaginal IgA and ADP don't differ significantly between groups overall, it is possible that the increase in activated target cells in the IM group is a key reason why the low Neut animals acquired quickly and if these were not increased in the ID group it could be a significant factor leading to reduced acquisition, especially when combined with the demonstrated differences in IgA and ADP responses.

We agree with the Reviewers' opinion and incorporated the same in the discussion section (lines. 377-385). Unfortunately, we do not have the privilege of sampling vaginal biopsies in vaccine protection studies prior to challenge to avoid potentially inducing damage or inflammation at the site of viral challenge.

8. When comparing acquisition in the animals with high and low INF γ + CD4 T cells the route of vaccine must be excluded and both IM and ID animals meeting the definitions included in order to achieve a statistically significant result. This suggests that in addition to the route of vaccine there are other factors such as background inflammation

at the time of vaccination or other animal intrinsic factors that must contribute to the generation of these responses and these factors should be evaluated if possible and discussed. In fact, in figure 4 it appears that the background inflammation may in fact be different between the two groups prior to receiving MVA (if D0 is collected prior to MVA vaccination) as the activated proinflammatory CD86+ classical monocytes are significantly higher (twice the level) in the IM group compared to the ID group at the baseline time point.

We agree with the Reviewer that it is possible multiple other factors such as background inflammation, composition of microbiome and the frequency of proliferating CD4 T cells in the mucosa could influence the protection outcome. However, with the group size we have, it is not possible to really address the impact of non-vaccine specific immune factors in this study especially within each vaccine group.

9. Why is there no basal data for the IM group on the GC-Tfh and GC-B cell graphs in fig 3?

Unfortunately, we didn't collect the FNAs samples from these animals at pre vaccination time point. Even though we don't have basal data for the IM group, it is clear from data post DNA vaccinations that they are comparable between the groups. We have removed basal data to avoid confusion and represented the range of GC-Tfh at baseline as shaded area on the graph. We clarified the same in the legend stating that basal data was from MVA-ID animals (lines 986-987).

10. The authors do a good job in Fig 6 of pointing out the potential for increased target cells as a potential factor in the differences in acquisition, but this may need to receive more attention in the text as a potential reason why small differences in levels of vaginal IgA and serum ADP, that are not significantly different between the IM/ID groups as a whole, lead to significant differences in acquisition based on the different routes.

We have updated the results section to highlight a more prominent role for increased target cell hypothesis as suggested by the Reviewer (lines 323-326).

Response to Reviewer #2:

The manuscript "Intradermal but not intramuscular MVA immunizations protect against intravaginal tier2 SHIV challenges in macaques" by Bollimpelli et al. pertains to how routes of administration of the modified vaccinia Ankara elicited unique protection profiles in macaques. The authors hypothesize that a particular route of vaccination "... could alter the magnitude and functional quality of the cellular and humoral immune response owing to the differences in the type and density of innate cells present in the skin and the muscle". The authors observed enhanced protection in the intra-dermally immunized group compared to the intra-muscular group and conduct a series of immunological studies to probe the correlates of immunity. These studies pointed to enhanced mucosal IgA and serum phagocytic activity, linked to enhanced Tfh responses and reduced inflammatory intermediate macrophage signatures *providing unique insights for next generation vaccine design*. This study

corroborates previously defined correlates of immunity observed across human and non-human primate studies and **points to a new strategy to tune antibody quality** that may be key to protection against HIV. There is a desperate need for enhanced multivariate and even machine learning analyses to help explain the relationship between the immune correlates both run in a discriminant as well regression mode. There are a few addressable issues listed below:

We are pleased to know that the Reviewer thinks our findings provide unique insights for next generation vaccine design and point to a new strategy to tune antibody quality. We are thankful to the Reviewer for these encouraging comments.

Major

1. The statistical grouping of viral loads in Figure 1C is not clear. The asterisks' presumably indicate the significance of that group (MVA-ID and MVA-IM) from controls. But are there additional differences within groups? The comparisons can be clarified with lines on the figures.

We have updated Figure 1C to include an additional significant comparison between MVA-ID and MVA-IM at week 3 post infection.

2. The authors find an association between neutralization in the IM group and mucosal IgA and peripheral ADP with viral control in the ID-MVA group, however whether these features are correlated, or explain protection orthogonally is unclear. A multivariate model could provide critical insight on the independent predictive accuracy of each feature in protection. It would be interesting to see if the signature of protection can be enriched if both arms are combined, with a model accounting for different immunization strategies.

We found an association between Vaginal IgA Vs serum ADP, and included it in SFig 4G. We didn't find any other associations between protection correlates. The additional correlations within antibody parameters are included in S.Fig4. We have also included the correlation analyses with both groups combined in Fig. 2E, 2H, 2I. When both groups are combined, p and r values are color coded in black (only significant correlations are mentioned). We also tried a multivariate analysis but this did not yield new insights into protection correlates. An additional correlation matrix showing the association between protection correlates and other cellular parameters is included in SFig. 7.

3. Along the same lines, the enhanced protection against infection in ID but potentially enhanced viral control observed with IM vaccination is quite interesting. Correlates of viral control are not actively investigated in this study. A similar multivariate model probing the correlates of viral control, with animal groups combined, could point to additional unique immune correlates of viral control.

We agree with the Reviewer that better viral control in the MVA-IM group is an interesting phenomenon. We tried to analyze all the parameters to find the possible immune correlates associated with this viral control. However, we could only identify serum IgG (negative association) and CD4-IFN gamma (positive association) as significant correlates with peak viral loads. We also did multivariate (PCA) analysis to find additional parameters but our analysis didn't yield additional observations.

4. For Figures 3A and 3B, the basal group representation as belonging to MVA-ID alone is confusing. The authors could greatly clarify these figures by removing the basal column and quantifying % changes or fold changes of MVA-ID and MVA-IM over basal.

- a. a % or fold increase in immune cell phenotypes from basal could provide enhanced clarity on differences across groups.

Unfortunately, we did not collect FNAs from the MVA-IM group animals at pre vaccination time point and because of this it will not be possible for us to calculate % changes or fold changes over baseline for the MVA-IM group. In addition, showing absolute values would allow the reader a better idea of the magnitude of the response. We agree with the Reviewer that showing baseline response only for one group can be confusing. To avoid this, we have removed the data for baseline collection and indicated the range for baseline values as horizontal dotted lines with shading on the graph.

5. The authors examine the differences in Th/Tfh and transcriptional profiles across the ID and IM arms but do not probe whether particular profiles or signatures explain IgA levels, ADP and neutralization.

As per reviewers' suggestion, we included a correlation matrix heat map with associations between protection correlates and other immune parameters analyzed in the current study. This is included in SFig. 7.

6. Looking at the manufacturer's protocols for the Ultra Low Input RNA kit, it is unclear if this is a bulk reverse transcription or an oligo-dT-based protocol. If this was a bulk reverse transcription, are there differences between the groups in non-coding RNAs? If this was an oligo-dT-based reverse transcription, then the authors need to clearly state that in their methods.

The kit uses oligo-DT based priming for the reverse transcription step, which would yield low recovery of non-coding RNAs. Therefore, we focused our analysis on protein-coding transcripts. We have now updated the methods to state this explicitly (lines 655-657).

7. The read counts standardizing for RNAseq need to be articulated more clearly. It would also help if the authors included a few housekeeping genes that remained constant between the two groups such as GAPDH, RPL19, etc.

Read counts standardizing for RNAseq is detailed further in RNAseq methodology. Rather than showing housekeeping genes, we determined the efficiency of normalization on the dataset as a whole using RLE plots. We have now added un-normalized and normalized RLE plots to the manuscript. (SFig. 10). The "Raw" plot shows the expression data for un-normalized values and the "Normalized" plot shows the expression data for normalized values.

Minor

1. The EM images in SFig1 and SFig 2 should be at the same magnification and require a scale bar.

We have magnified sFig1 EM image to match the magnification of EM image in SFig 2 with scale bars.

2. Were some of the blots in SFig 1B run on the same gel? It looks like the lanes of the ladder do not correspond to the same gel, which is fine if they were run at the same time. But it should be disclosed.

CD40L blots for lysate and supernatant were run on different gels but at the same time. This is disclosed in the legend.

3. There is no reference to panel A in the Figure 2 legend.

Corrected.

4. The black and blue dots in Figure 2E, H, and I, and Figure 3G-I are difficult to discern. Please consider using a lighter shade of blue, or using gray instead of black.

We did not use any black color dots in those figures. They have only red and blue dots. We excluded data from controls (represented with black color) for post vaccine correlates.

5. The legend font in Figures 3A, 3B, and 3E is extremely small and borderline non-legible

Corrected accordingly.

6. The titles of the graphs in SFig 4 are cutoff.

Corrected.

7. SFig 6B and SFig 6C need to be incorporated into the Main Figures. They are far too important to the authors' conclusions to be the final two panels in the supplemental figures.

We have moved SFig. 6B and SFig. 6C to Fig. 2 as Fig. 2I and Fig2. J as suggested by the Reviewer.

8. Articles are missing throughout the manuscript and make it difficult to read.

We apologize for this error. We carefully read through the manuscript and corrected as much as possible.

Response to Reviewer #3:

The authors previously demonstrated that 1) protein-only vaccination protects from autologous challenge in rhesus macaques and that protection is associated with neutralizing titer above 1:300 and 2) when combined with heterologous prime-boost with viral vector vaccines, some animals with neutralizing antibody (NAb) titers below 1 :300 are protected (Arunachalam et al, 2020). Here the authors report with another heterologous prime-boost regimen, that the route of administration of MVA-vectorized vaccine matters for protection. When MVA is administered intradermally (MVA-ID),

macaques are protected against the acquisition of infection, whereas they are not when MVA is injected intramuscularly (MVA-IM). However, MVA-IM-vaccinated animals exhibit a better control of viral replication.

Interestingly, similar though heterogeneous NAb titers are detected in both groups (ranging from below detection limit to 1:6,000) and in both groups, titers above 1:50 correlate with protection. In the MVA-ID group only, some animals with neutralizing titers below 1:50 are protected and protection is associated with Ab-dependant phagocytic activities (ADP) in serum and with vaginal IgA binding titers. MVA-ID immunization is also associated with higher germinal center Tfh and B-cell responses, and in blood, with a lower ratio of Th1 to Tfh cell activation and a lower activation of intermediate monocytes and inflammasome.

Overall, ***the data are of interest to the HIV vaccine field*** and more largely to the vaccine field, emphasizing the need of testing more in depth the impact of immunization route. Results are quite clearly presented and the manuscript is most of the time well written. However, language and style can be improved in some sections (see minor issues) and some revisions are needed for publication.

We thank the Reviewer for encouraging comments. We have carefully read the manuscript and tried to improve on the language and style as much as possible.

Major concerns

1. The title, the abstract and the end of the introduction (lines 96-97) emphasize the difference of protection between MVA-ID and MVA-IM groups. Actually, MVA-ID group is protected against the acquisition of infection and both groups are protected against high viral loads in breakthrough infection, with MVA-IM doing even better than MVA-ID (Fig. 1). The authors should reformulate to align with the message of the conclusion section (line 374 to 377), which is clearer and more in line with the data.

We agree with the Reviewer. Unfortunately, we are limited by the word count in the title and abstract. We have updated Fig. 1C and the introduction, results and discussion sections to clearly state that viral control was better in the MVA-IM group (lines 99-101, 131-138, 342-344).

2. The correlates of protection against high viral load should also be analyzed comprehensively (see below).

We tried to analyze all the parameters to find the possible immune correlates associated with this viral control. However, we could only identify serum IgG (negative association) and CD4-IFN gamma (positive association) as significant correlates with peak viral loads. We also did multivariate (PCA) analysis to find additional parameters but our analysis didn't yield additional observations.

3. The Control group seems to be the same as in Arunachalam et al, 2020. This should be indicated in the Materials & Methods Animals section.

We indicated the same as suggested (lines 451-453).

4. The authors should indicate if all animals were challenged simultaneously in the

Materials & Methods Animals section. If not, the authors should precise to which cohort belong each animal in Sup Table 1.

Animals were challenged in two cohorts and we have now included the cohort numbers in Sup Table 1.

5. The animal groups are quite heterogeneous in term of age: Control group median 3 years range 12, MVA-IM group median 6 years range 4, MVA-ID group median 7.5 years range 9. The author should indicate whether there is a significant difference and if it has an impact on the results, particularly if the protection outcome is different according to the age.

Animal age was comparable between the vaccinated groups. However vaccinated animals were significantly older compared to control. However, we did not observe a significant correlation (non-parametric) between the age and viral acquisition either in the control or vaccine groups. In our previous study we observed a poorer protection in older animals compared to younger animals (C. Petitdemange et al., 2019; JCI insight; [10.1172/jci.insight.126047](https://doi.org/10.1172/jci.insight.126047)).

6. As the age are quite different, the authors should add the animal weights in Sup Table 1.

We have added animal weights to the Sup Table.1.

7. The authors should provide details on the randomisation method they used.

Randomization was only done for vaccinated animals since the controls were added later on. We used the age and weight of the animals for randomization (lines 450-451)

8. Line 100 : the authors should precise that the protection that is observed in MVA-ID and not in MVA-IM is against the acquisition of infection

Changed as per the Reviewers' suggestion.

9. Figure 1

a. Title : the authors should precise that MVA-ID protects against the acquisition of infection

Changed as per the Reviewers' suggestion.

b. B. Though the Control group is the same as in Arunachalam et al, 2020, the survival curve is slightly different : the authors should correct or explain

We thank the Reviewer for pointing this out. We realized that we made an error for animal in our 2020 paper. The data presented in this manuscript is accurate. However, this error in 2020 paper doesn't change any P values and conclusions. We will submit a correction to 2020 paper.

c. B. The authors should provide calculation details for vaccine efficacy per challenge in the legend or in the Mat&Meth section

The vaccine efficacy was calculated based on the method by Hudgens and Gilbert (Assessing vaccine effects in repeated low-dose challenge experiments. *Biometrics* 2009, 65, 1223-1232). This is included in methodology section (Lines 511-514).

d. C. The authors should add a 5th graph, with statistical analysis, to compare the areas under the curve of viral RNA kinetics from week 0 to week 11 for the three groups

A graph comparing the area under the curve of viral loads from infected animals has been included accordingly in Fig 1D. We expanded discussion on viral control in a separate paragraph.

e. C. Kruskal-Wallis test should be performed when more than 2 groups are compared before comparing groups using Mann-Whitney test. They authors perform this test and mention it in the legend and in the Statistical analysis section accordingly.

Kruskal-Wallis test was performed before Mann-Whitney test for comparisons involving more than two groups. The same has been mentioned in methods section (Lines 680-682).

f. C. The authors should check for consistency or clarify: in the legend, week 1 is defined as the first detection of positive viral load, which is fine, whereas my understanding from line 440-441 is that it is week 2.

As the Reviewer might be aware, the first detection of viremia in plasma is little delayed following the intravaginal challenge compared to intrarectal challenge. In our experience during titration of viruses in NHPs where we give a gap of 3 weeks between each exposure, we generally see viremia at one week post rectal challenge but at two weeks post vaginal challenge in >90% instances. Because of this we considered 2 weeks before the first viral load detection as the challenge number at which animal was productively infection.

As pointed out by the Reviewer, for plotting viral load post infection we depicted as week 1 post infection but in reality, it is more likely to be week 2 post infection. To avoid this confusion, we have changed the x-axis legend in Fig1 viral load graphs to 'week post first detection of viremia'.

10. The authors should provide a supplementary figure with the viral load curves of each animal, with animal IDs, according to the time of the first challenge. This will give insight on whether viral replication is different when animals are infected at the first challenges or later on.

We have added this figure (SFig.3). We do not see any significant influence on virus replication with respect to the number of challenges at which an animal was infected.

11. The analysis of the correlates of protection is quite confusing.

a. It is not clear enough whether all animal with neut > 50 were protected (lines 158-160)

Of the six animals with neut > 50, five were completely protected i.e never showed any viral load even after 10 challenges. The other one animal became infected at the 10th challenge. We have modified the text to clarify this (Lines 174-176).

b. The authors should not analyze the MVA-ID animals that got infected after 9 challenges along with the protected ones as the delay of infection can be stochastic especially since one animal in the Control group got infected after 9 challenges too.

As per the Reviewers' suggestion, we avoided combining animals that got infected after the 9th challenge with protected animals. We have modified SFig 3H (now SFig 4H) accordingly.

c. Line 161 : the difference is stated as significant whereas it is not the case in Fig 2F.

Statistical significance between the vaccinated groups is significant ($p=0.023$). However, the difference between MVA-ID and control animals was approaching significance ($p=0.056$). We have updated the figure to include the p values between the vaccine groups.

d. As there is a delay of 10 weeks between the first and the last challenge, the Neut data at the time of infecting challenge for the breakthrough animals and at the end of the 10 challenges for the protected ones would be of high interest to explore 1) the durability of the response and 2) the impact of repeated vaginal exposures to vaccine-induced responses.

We agree with the Reviewer. Unfortunately, we do not have data on neutralization titers at the time of infecting challenge for the breakthrough animals and at the end of the 10 challenges for the protected animals.

e. Figure 2E :

i. There should be only one data plotted above the 10th challenge as there is only one animal that got infected at this challenge ; the same for Fig. 2H and 2I.

We showed uninfected animals at the 10th challenge and this caused this confusion. We have now separated the uninfected animals and plotted them as UI after the 10th challenge.

ii. The data of animals that remained protected from SHIV acquisition should not be plotted at all in this graph ; the same for Fig. 2H and 2I.

We respectfully disagree. Since these animals resisted 10 challenges and we were correlating with the number of challenges to infection, it is important to keep these animals on the correlation graphs in 2E, 2H (currently 2K) and 2I (currently 2L). Hope the Reviewer can understand.

iii. The threshold line should be plotted at 50 and not at 20 as 50 is the value chosen by the authors to discriminate between low and high Neut. This should also be precised in the legend.

The threshold line has been updated to 50 and the same is mentioned in the legend.

f. Figure 2F : the authors should add the curves of the high neut animals, for each group
Included the survival curves of high neut animals.

g. Please analyze the correlates of protection with the same type of graph as Sup Fig 3H, but discriminating only by « infected » vs « protected », along with Mann-Whitney test, for all the parameters that were studied.

As suggested by the Reviewer we analyzed all the parameters comparing protected and infected using the Mann-Whitney test. Only Neutralization, Vaginal IgA and ADP showed statistical significance between protected and infected and hence included them in Sup Fig4.

h. Please provide the analysis of the correlates of protection against high viral load as in Sup. Fig. 5A, for all the parameters that were studied. It is only provided for binding BG505-specific Ab (line 176) and it would be of high interest for the other ones (neut, mucosal Ab titers etc)

As mentioned above, we tried to analyze all the parameters to find the possible immune correlates associated with this viral control. However, we could only identify serum IgG (negative association) and CD4-IFN gamma (positive association) as correlates with peak viral loads of infected animals. We also did multivariate (PCA) analysis to find additional parameters but our analysis didn't yield any additional observations.

i. For improved clarity, the Neut values at the time of the 1st challenge should be added in the Sup Table 1.

Done as suggested.

j. I would suggest to represent the correlates of protection and their associations by a correlation matrix heatmap with Ab and cellular parameters in order to have a global overview.

As per the Reviewers' suggestion, we included a correlation matrix heat map with associations between protection correlates and other cellular parameters analyzed in the current study. This is included in SFig. 7.

12. Lines 309-311 : the authors should precise that the protection they are talking about is against the acquisition of infection.

Corrected accordingly in the discussion as mentioned by the Reviewer.

13. The results should be discussed in light of two recent publications that studied the impact of the route of MVA immunization in macaques (Rosenbaum et al, 2021, <https://doi.org/10.3389/fimmu.2021.645210> and Feraoun et al, 2022, <https://doi.org/10.3389/fimmu.2021.784813>).

We thank the Reviewer for providing these important references. We have discussed these findings in the discussion section (Lines 401-406, 440-443).

Minor issues

14. Please clarify the role of CD40L in the DNA vaccine

We have updated the results section as suggested by the Reviewer (lines 111-113).

15. References 5 and 6 are the same

Corrected.

16. Lines 151-153 : « On the day of first challenge » ; then check for the wording which seems odd or remove « ID50 is less than 1 :20 »

Removed « ID50 is less than 1 :20 ».

17. Line 159 : check for odd wording or remove « were »

Removed.

18. Line 185-187 : please simplify the wording ; I would suggest « The frequency of germinal center T follicular helper cells (GC-Tfh) ranged from 1-4% (average 2%) at baseline and remained unchanged after the two DNA vaccinations »

Modified as suggested.

19. Line 200 : than in the MVA-IM group

Corrected.

20. Fig. 2

a. Legend line 853: « A) » is missing

Corrected.

b. Graph D : title of the X « two weeks post immunization » is not convenient for the last data plotted which are the data of the day of challenge. Please correct by separating the X axe for instance

Corrected.

c. Graph H : typo in the title

Corrected.

21. Fig Sup. 3 : add the group color code in the figure for improved clarity

Color code has been added.

22. Typos

a. Check abbreviations: liter is L and not l.

Corrected

b. vs : italicize

Corrected.

c. Add a space between number and unit, i.e. line 424 « 760 µg » (many occurrences)

Corrected.

d. Line 425 and others: correct « 1x10⁸ PFU » (several occurrences)

Corrected.

REVIEWERS' COMMENTS

Reviewer #1 (Remarks to the Author):

The authors have responded to all my concerns.

Reviewer #3 (Remarks to the Author):

The authors have addressed my comments satisfactorily.
Congratulation for your great work.

For completeness and clarity : I have been asked to provide additional comment on the response of the authors to the prior concerns of reviewer 2. I reviewed the comments of Reviewer 2 and the answers the authors made to them. My conclusion is that the authors sufficiently addressed Reviewer 2's concerns.